# Distribution Backtracking Builds A Faster Convergence Trajectory for Diffusion Distillation

**Shengyuan Zhang**[1], **Ling Yang**[3], **Zejian Li**[2*], **An Zhao**[1], **Chenye Meng**[1], **Changyuan Yang**[4],
**Guang Yang**[4], **Zhiyuan Yang**[4], **Lingyun Sun**[1]

[1]College of Computer Science and Technology, Zhejiang University

[2]School of Software Technology, Zhejiang University

[3]Peking University

[4]Alibaba Group

[1,2]{zhangshengyuan,zejianlee,zhaoan040113,mengcy,sunly}@zju.edu.cn

[3]{yangling0818}@163.com

[4]{changyuan.yangcy,qingyun,adam.yzy}@alibaba-inc.com

∗ Corresponding author

## Abstract

Accelerating the sampling speed of diffusion models remains a significant challenge. Recent score distillation methods distill a heavy teacher model into a student generator to achieve one-step generation, which is optimized by calculating the difference between two score functions on the samples generated by the student model. However, there is a score mismatch issue in the early stage of the score distillation process, since existing methods mainly focus on using the endpoint of pre-trained diffusion models as teacher models, overlooking the importance of the convergence trajectory between the student generator and the teacher model. To address this issue, we extend the score distillation process by introducing the entire convergence trajectory of the teacher model and propose **Dis**tribution **Back**tracking Distillation (**DisBack**). DisBask is composed of two stages: *Degradation Recording* and *Distribution Backtracking*. *Degradation Recording* is designed to obtain the convergence trajectory by recording the degradation path from the pre-trained teacher model to the untrained student generator. The degradation path implicitly represents the intermediate distributions between the teacher and the student, and its reverse can be viewed as the convergence trajectory from the student generator to the teacher model. Then *Distribution Backtracking* trains the student generator to backtrack the intermediate distributions along the path to approximate the convergence trajectory of the teacher model. Extensive experiments show that DisBack achieves faster and better convergence than the existing distillation method and achieves comparable or better generation performance, with an FID score of 1.38 on the ImageNet 64×64 dataset. DisBack is easy to implement and can be generalized to existing distillation methods to boost performance. Our code is publicly available on https://github.com/SYZhang0805/DisBack.

## 1 Introduction

Recently, generative models have demonstrated remarkable performance across diverse domains such as images (Kou et al., 2023; Yin et al., 2024a), audio (Evans et al., 2024; Xing et al., 2024), and videos (Wang et al., 2024; Chen et al., 2024). However, existing models still grapple with the "trilemma" problem, wherein they struggle to simultaneously achieve high generation quality, fast generation speed, and high sample diversity (Xiao et al., 2021). Generative Adversarial Networks (GANs) (Goodfellow et al., 2014) can rapidly produce high-quality samples but often face mode collapse issues. Variational Autoencoders (VAEs) (Kingma & Welling, 2014) offer stable training but tend to yield lower-quality samples. Recently, Diffusion models (DMs) have emerged as a

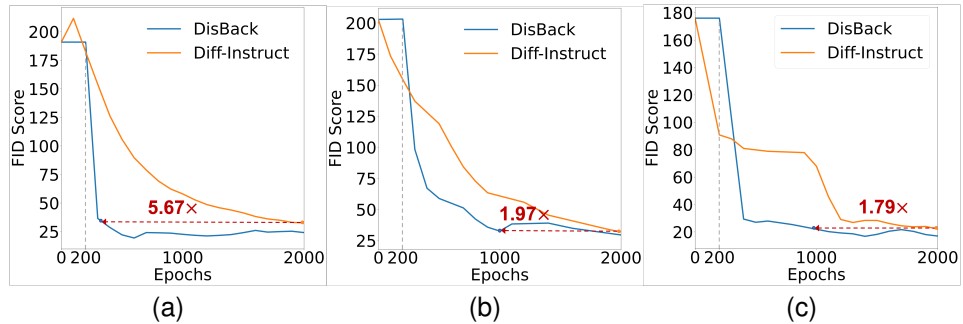

Figure 1: The comparison of the distillation process between existing SOTA score distillation method Diff-Instruct (Luo et al., 2023c) and proposed DisBack on (a) CIFAR10, (b) FFHQ 64x64, and (c) ImageNet 64x64 datasets. The first 200 epochs refer to the computational overhead of the degradation recording stage of DisBack. DisBack achieves a faster convergence speed due to the constraint of the entire convergence trajectory between the student generator and the teacher model.

competitive contender in the generative model landscape (Fan et al., 2023; Zhou et al., 2023; Xu et al., 2024). Diffusion models can generate high-quality, diverse samples but still suffer from slow sampling speeds due to iterative network evaluations.

To accelerate the sampling speed, the score distillation method tries to distill a heavy teacher model to a student generator to reduce the sampling cost and achieve the one-step generation (Bao et al., 2023; Luo et al., 2023c; Yin et al., 2024b). The score distillation method optimizes the student generator by calculating the difference between two score functions on the samples generated by the student generator. However, as the generated distribution is far from the training distribution at the beginning, the generated sample lies outside the training data distribution. Thus, the predicted score of the generated sample from the teacher model does not match the sample's real score in the training distribution. This mismatch issue is reflected by unreliable network predictions of the teacher model, which prevents the student model from receiving accurate guidance and leads to a decline in final generative performance. We identified that this issue arises because existing score distillation methods mainly focus on using the endpoint of the pre-trained diffusion model as the teacher model, overlooking the importance of the convergence trajectory between the student generator and the teacher model. Without the constraint of the convergence trajectory, the mismatch issue causes the student generator to deviate from a reasonable optimization path during training, leading to convergence to suboptimal solutions and a decline in final performance.

To address this problem, we extend the score distillation process by introducing the entire convergence trajectory of the teacher model and propose **Dis**tribution **Back**tracking Distillation (**DisBack**) for a faster and more efficient distillation. The construction of DisBack is based on the following insights. In practice, the convergence trajectory of most teacher models is inaccessible, particularly for large models like Stable Diffusion (Rombach et al., 2022). Because the trajectory of distribution changes is bidirectional, it is possible to construct a degradation path from the teacher model to the initial student generator, and the reverse of this path can be viewed as the convergence trajectory of the teacher model. Compared with fitting the teacher model directly, fitting intermediate targets along the convergence trajectory can mitigate the mismatch issue. Thus, the DisBack incorporates degradation recording and distribution backtracking stages. In the degradation recording stage, the teacher model is tuned to fit the distribution of the initial student generator and obtains a distribution degradation path. The path includes a series of in-between diffusion models to represent the intermediate distributions of the teacher model implicitly. In the distribution backtracking stage, the degradation path is reversed and viewed as the convergence trajectory. Then the student generator is trained to backtrack the intermediate distributions along the path to optimize towards the convergence trajectory of the teacher model. In practice, the degradation recording stage typically requires only a few hundred iterations. Therefore, the proposed method incurs trivial additional computational costs. Compared to the existing score distillation method, DisBack exhibits a significantly increased convergence speed (Fig. 1), and it also delivers superior generation performance (Fig. 2).

Our main contributions are summarized as follows. (1) We extend the score distillation process by introducing the constraint of the entire convergence trajectory of the teacher model and propose Dis-

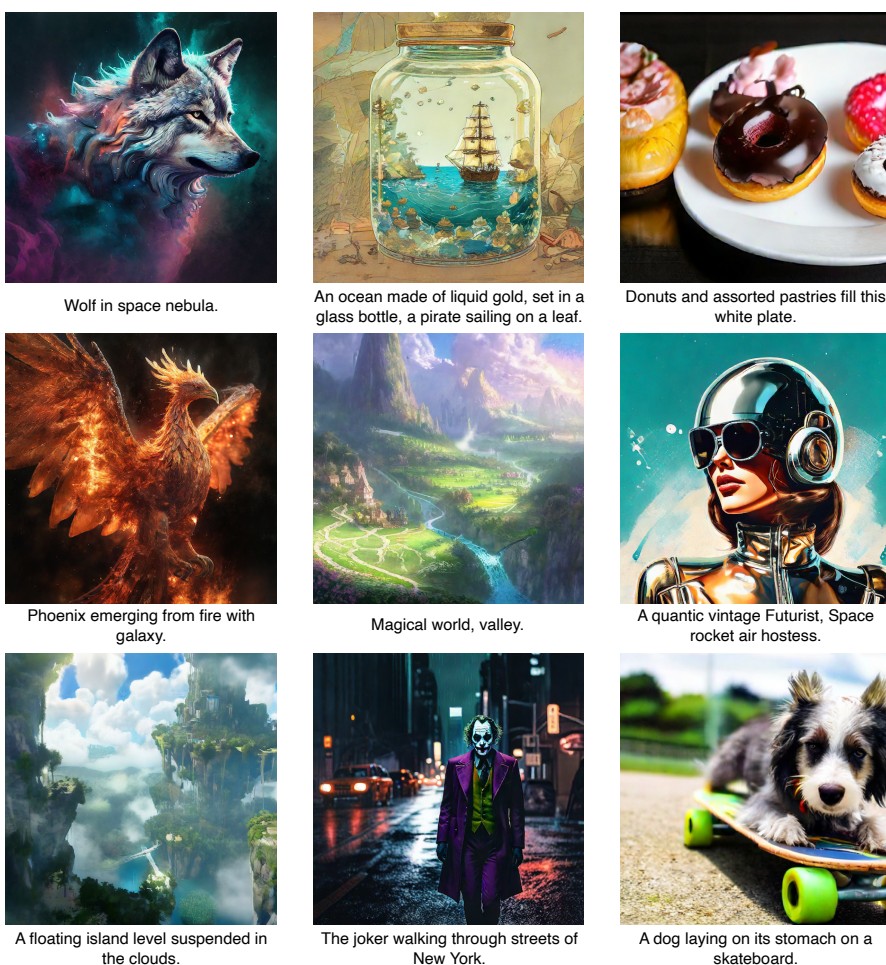

Figure 2: Several examples of 1024×1024 images generated by our proposed one-step DisBack model distilled from SDXL (Podell et al., 2024).

tribution Backtracking Distillation (DisBack), which achieves a faster and more efficient distillation (Sec. 4). (2) Extensive experiments demonstrate that the proposed DisBack accelerates the convergence speed of the score distillation process while achieving comparable or better generation quality compared to existing methods (Sec. 5). (3) The contribution of DisBack is orthogonal to those of other distillation methods. Researchers are encouraged to incorporate our DisBack training strategy into their distillation methods.

## 2 RELATED WORKS

**Efficient diffusion models.** To improve the efficiency of the diffusion model, existing methods use knowledge distillation to distill a large teacher model to a small and efficient student model (Yang et al., 2022). The progressive distillation (Salimans & Ho, 2021) progressively distills the entire sampling process into a new diffusion model with half the number of steps iteratively. Building on this, the classifier-guided distillation (Sun et al., 2023) introduces a dataset-independent classifier to focus the student on the crucial features to enhance the distillation process. Guided-distillation (Meng et al., 2023) proposes a classifier-free guiding framework to avoid the computational cost of additional classifiers and achieve high-quality sampling in only 2-4 steps. Recently, the Consistency Model (Song et al., 2023) uses the self-consistency of the ODE process to achieve one-step distillation, but this is at the expense of generation quality. To mitigate the surface of the sample quality caused by the acceleration, the Consistency Trajectory Model (Kim et al., 2024a) combines the adversarial training and denoising score matching loss to further improve the performance. Latent

Adversarial Diffusion Distillation (Sauer et al., 2024) leverages generative features from pre-trained latent diffusion models to achieve high-resolution, multi-aspect ratio, few-step image generation.

**Score distillation for one-step generation.** Diff-Instruct (Luo et al., 2023c) distills a pre-trained diffusion model into a one-step generator by optimizing it with the gradient of the difference between two score functions—one for the pre-trained diffusion distribution and the other for the generated distribution. Adversarial Score Distillation (Wei et al., 2024) further employs the paradigm of WGAN and retains an optimizable discriminator to improve performance. Additionally, Swiftbrush (Nguyen & Tran, 2024) leverages score distillation to distill a Stable Diffusion v2.1 into a one-step generator and achieve competitive results. DMD (Yin et al., 2024b) suggests the inclusion of a regression loss between noisy images and corresponding outputs to alleviate instability in the distillation process in text-to-image generation tasks. DMD2 (Yin et al., 2024a) introduces a two-time-scale update rule and an additional GAN loss to address the issue of generation quality being limited by the teacher model in DMD, achieving superior performance. Recently, HyperSD (Ren et al., 2024) integrates score distillation with trajectory segmented consistency distillation and human feedback learning, which achieves SOTA performance from 1 to 8 inference steps.

**Mismatch issues in diffusion model.** Existing studies have already discussed the issue of score mismatch in diffusion models Chao et al. (2022); Kim et al. (2024b). DLSM Chao et al. (2022) deeply analyzed the mismatch between the posterior score estimated by diffusion models and the true likelihood score in conditional generation and proposed the Denoising Likelihood Score Matching loss to train classifiers for more accurate conditional score estimation. TIW-DSM Kim et al. (2024b) highlighted that dataset biases cause mismatches between the generator's and true data distributions, introducing time-varying importance weights and score correction terms to mitigate this issue by assigning different weights to data points at each time step and correcting the scores. This paper focuses on the mismatch arising from the disparity between the initial generator's distribution and the teacher model's training distribution, causing discrepancies between predicted and true scores.

## 3 Preliminary

In this part, we briefly introduce the score distillation approach. Let $q_0^G$ and $q_t^G$ be the distribution of the student generator $G_{stu}$ and its noisy distribution at timestep $t$. In addition, $q_0$ and $q_t$ are the training distribution and its noisy distribution at timestep $t$. By optimizing the KL divergence in Eq. (1), we can train a student generator to enable one-step generation (Wang et al., 2023).

$$\min_{\eta} D_{KL}\left(q_0^G\left(\boldsymbol{x}_0\right)\|q_0\left(\boldsymbol{x}_0\right)\right) \tag{1}$$

Here $\boldsymbol{x}_0 = G_{stu}(\boldsymbol{z};\eta)$ is the generated samples, and $\eta$ is the trainable parameter of $G_{stu}$. However, due to the complexity of $q_0$ and its sparsity in high-density regions, directly solving Eq.(1) is challenging (Song & Ermon, 2019). Inspired by Variational Score Distillation (VSD) (Wang et al., 2023), Eq.(1) can be extended to optimization problems at different timesteps $t$ in Eq. (2). As $t$ increases, the diffusion distribution becomes closer to a Gaussian distribution.

$$\min_{\eta} \mathbb{E}_{t\sim\mathcal{U}(0,1),\boldsymbol{\epsilon}\sim\mathcal{N}(0,I)} D_{KL}\left(q_t^G\left(\boldsymbol{x}_t\right)\|q_t\left(\boldsymbol{x}_t\right)\right) \tag{2}$$

Here $\boldsymbol{x}_t$ is the noisy data and $\boldsymbol{x}_t \mid \boldsymbol{x}_0 \sim \mathcal{N}(\boldsymbol{x}_0,\sigma_t^2 I)$. Theorem 1 proves that introducing the additional KL-divergence for $t > 0$ does not affect the global optimum of Eq.(1)

**Theorem 1 (The global optimum of training (Wang et al., 2023))** *Given $t > 0$, we have,*

$$D_{\mathrm{KL}}\left(q_t^G\left(\boldsymbol{x}_t\right)\|q_t\left(\boldsymbol{x}_t\right)\right) = 0 \Leftrightarrow D_{\mathrm{KL}}\left(q_0^G\left(\boldsymbol{x}_0\right)\|q_0\left(\boldsymbol{x}_0\right)\right) = 0 \tag{3}$$

Therefore, by minimizing the KL divergence in Eq. (2), the student generator can be optimized through the following gradients:

$$\nabla_{\eta} D_{\mathrm{KL}}\left(q_t^G\left(\boldsymbol{x}_t\right)\|q_t\left(\boldsymbol{x}_t\right)\right) = \mathbb{E}_{t,\boldsymbol{\epsilon}}\left[\left[\nabla_{\boldsymbol{x}_t}\log q_t^G\left(\boldsymbol{x}_t\right) - \nabla_{\boldsymbol{x}_t}\log q_t\left(\boldsymbol{x}_t\right)\right]\frac{\partial \boldsymbol{x}_t}{\partial \eta}\right] \tag{4}$$

Here the score of perturbed training data $\nabla_{\boldsymbol{x}_t}\log q_t\left(\boldsymbol{x}_t\right)$ can be approximated by a pre-trained diffusion model $s_\theta$. The score of perturbed generated data $\nabla_{\boldsymbol{x}_t}\log q_t^G\left(\boldsymbol{x}_t\right)$ is estimated by another

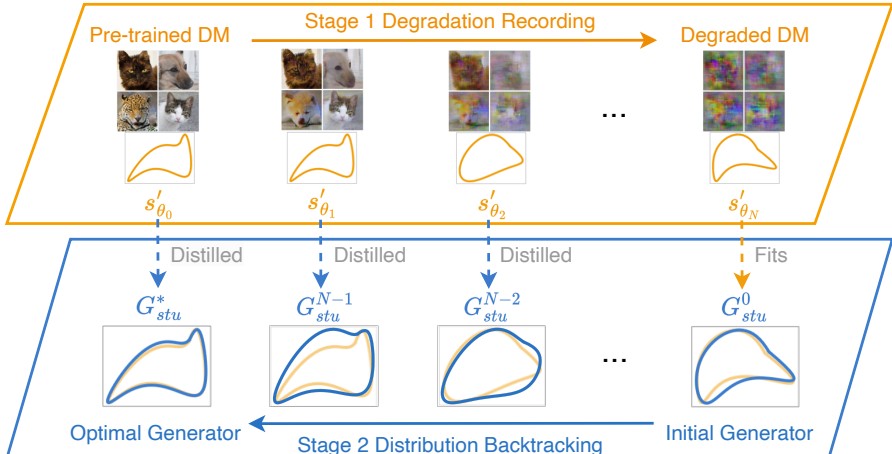

Figure 3: The overall framework of DisBack. Stage 1: An auxiliary diffusion model is initialized with the teacher model $s_\theta$ and then fits the distribution of the initial student generator $G^0_{stu}$. The intermediate checkpoints $\{s'_{\theta_i} \mid i = 0, \ldots, N\}$ are saved to form a degradation path. The degradation path is then reversed and viewed as the convergence trajectory. Stage 2: The intermediate node $s_{\theta_i}$ along the convergence trajectory is distilled to the student generator sequentially until the generator converges to the distribution of the teacher model.

diffusion model $s_\phi$, which is optimized by score matching with generated data (Song et al., 2021b):

$$\min_\phi \mathbb{E}_{t,\epsilon} \left\| s_\phi\left(\boldsymbol{x}_t, t\right) - \frac{\boldsymbol{x}_0 - \boldsymbol{x}_t}{\sigma_t^2} \right\|_2^2 \tag{5}$$

Thus, the gradient of student generator in Eq.(4) is estimated as

$$\nabla_\eta D_{\mathrm{KL}}\left(q_t^G\left(\boldsymbol{x}_t\right) \| q_t\left(\boldsymbol{x}_t\right)\right) \approx \mathbb{E}_{t,\epsilon}\left[\left[s_\phi\left(\boldsymbol{x}_t, t\right) - s_\theta\left(\boldsymbol{x}_t, t\right)\right] \frac{\partial \boldsymbol{x}_t}{\partial \eta}\right] \tag{6}$$

The distribution of the student generator changes after its update. Therefore, $s_\phi$ also needs to be optimized based on the newly generated images to ensure the timely approximation of the generated distribution. Thus, the student generator and $s_\phi$ are optimized alternately.

In practice, $s_\phi$ has three initialization strategies: (1) $s_\phi$ is randomly initialized (Franceschi et al., 2023). (2) $s_\phi$ is initialized as $s_\theta$ or its LoRA (Hu et al., 2021; Wei et al., 2024). (3) $s_\phi$ is initialized by fitting the generated samples of student generator (Luo et al., 2023c). Beyond unconditional image generation (Ye & Liu, 2023), this method has also been applied to tasks such as text-to-image and image-to-image generation across various structures (Yin et al., 2024b; Hertz et al., 2023).

## 4 METHOD

### 4.1 INSIGHT

In this section, we introduce the **Dis**tribution **Back**tracking Distillation (**DisBack**). The key insight behind DisBack is the importance of the convergence trajectory. As mentioned in Sec.3, there are two score functions in score distillation, one representing the pre-trained diffusion distribution and the other representing the generated distribution. The student model is optimized using the gradient of the difference between these two score functions. Existing methods (Luo et al., 2023c; Yin et al., 2024b;a) directly use the endpoint of the pre-trained diffusion model as the teacher model, overlooking the intermediate convergence trajectory between the student generator and the teacher model. The resulting score mismatch issue between the predicted scores of the generated sample from the teacher model and the real scores causes the student model to receive inaccurate guidance. It ultimately leads to a decline in final performance. Constraining the convergence trajectory between the student generator and the teacher model during the distillation process can mitigate the

mismatch issue and help the student generator approximate the convergence trajectory of teacher models to achieve faster convergence. In practice, it is infeasible to obtain the convergence trajectory of most teacher models, especially for large models such as Stable Diffusion (Rombach et al., 2022). Reversely, it is possible to obtain the degradation path from the teacher model to the initial student generator. The reverse of this degradation path can be viewed as the convergence trajectory of the teacher model. Based on the above insights, we structure the proposed DisBack in two stages including the degradation recording stage and the distribution backtracking stage (Fig. 3).

## 4.2 DEGRADATION RECORDING

This stage aims to obtain the degradation path from the teacher to the initial student generator. The degradation path is then reversed and viewed as the convergence trajectory of the teacher model. The teacher here is the pre-trained diffusion model $s_\theta$ and the student is represented by $G_{stu}^0$.

Let $s_\theta'$ be a diffusion model initialized with the teacher model $s_\theta$, and it is trained on generated samples to fit the initial student generator's distribution $q_0^G$ with Eq. (7). By saving the multiple intermediate checkpoints during the training, we can obtain a series of diffusion models $\{s_{\theta_i}' \mid i = 0, \ldots, N\}$, where $s_{\theta_0}' = s_\theta \approx q_0$ and $s_{\theta_N}' \approx q_0^G$. These diffusion models describe the scores of non-existent distributions on the path, recording how the training distribution $q_0$ degrades to the initial generated distribution $q_0^G$. Algorithm 1 shows the process of obtaining the degradation path. Since distribution degradation is easily achievable, the degradation recording stage only needs trivial additional computational resources (200 iterations in most cases).

$$\min_\theta \mathbb{E}_{t, \epsilon} \left\| s_\theta' \left( \boldsymbol{x}_t, t \right) - \frac{\boldsymbol{x}_0 - \boldsymbol{x}_t}{\sigma_t^2} \right\|_2^2 \tag{7}$$

---

**Algorithm 1** Degradation Recording.

**Input:** Initial student generator $G_{stu}^0$ and pre-trained diffusion model $s_\theta$.

**Output:** Degradation path checkpoints $\{s_{\theta_i}' \mid i = 0, \ldots, N\}$
$s_\theta' \leftarrow s_\theta$

**while** not converge **do**
$\quad \boldsymbol{x}_0 = G_{stu}^0(\boldsymbol{z}; \eta)$

$\quad$ Update $\theta$ with gradient
$\quad \quad \frac{\partial}{\partial \theta_i} \mathbb{E}_{t, \epsilon} \left\| s_\theta' \left( \boldsymbol{x}_t, t \right) - \frac{\boldsymbol{x}_0 - \boldsymbol{x}_t}{\sigma_t^2} \right\|_2^2$
$\quad$ Save intermediate checkpoints $s_{\theta_i}'$
**end while**

---

**Algorithm 2** Distribution Backtracking.

**Input:** Initial student generator $G_{stu}^0$ and reverse path checkpoints $\{s_{\theta_i}' \mid i = N, \ldots, 0\}$

**Output:** One-step generator $G_{stu}^*$
$s_\phi \leftarrow s_{\theta_N}'$
**for** $i \leftarrow N - 1$ to $0$ **do**
$\quad$ **while** not converge **do**
$\quad \quad \boldsymbol{x}_0 = G_{stu}^0(\boldsymbol{z}; \eta)$

$\quad \quad$ Update $\eta$ with gradient
$\quad \quad \quad \mathbb{E}_{t, \epsilon} \left[ s_\phi \left( \boldsymbol{x}_t, t \right) - s_{\theta_i}' \left( \boldsymbol{x}_t, t \right) \right] \frac{\partial \boldsymbol{x}_t}{\partial \eta}$
$\quad \quad$ Update $\phi$ with gradient
$\quad \quad \quad \frac{\partial}{\partial \phi} \mathbb{E}_{t, \epsilon} \left\| s_\phi \left( \boldsymbol{x}_t, t \right) - \frac{\boldsymbol{x}_0 - \boldsymbol{x}_t}{\sigma_t^2} \right\|_2^2$
$\quad$ **end while**
**end for**

---

## 4.3 DISTRIBUTION BACKTRACKING

Given the degradation path from the teacher model to the initial student generator, the reverse path is viewed as a representation of the convergence trajectory between the initial student generator $G_{stu}^0$ and the teacher model $s_\theta$. The key to the distribution backtracking is to sequentially distill checkpoints in the convergence trajectory into the student generator. The last node $s_{\theta_N}'$ in the path is close to the initially generated distribution $q_0^G$. Therefore, in the distribution backtracking stage, we use $s_{\theta_{N-1}}'$ as the first target to distill the student generator. When near convergence, we switch the target to $s_{\theta_{N-2}}'$. The checkpoints $s_{\theta_i}'$ is sequentially distilled to $G_{stu}$ until the final target $s_{\theta_0}'$ is reached. During the distillation, the gradient of $G_{stu}$ is:

$$\text{Grad}(\eta) = \mathbb{E}_{t, \epsilon} \left[ \left[ s_\phi \left( \boldsymbol{x}_t, t \right) - s_{\theta_i}' \left( \boldsymbol{x}_t, t \right) \right] \frac{\partial \boldsymbol{x}_t}{\partial \eta} \right] \tag{8}$$

Table 1: The unconditional generation performance of DisBack. The FID ($\downarrow$) scores are shown.

| Model | NFE ($\downarrow$) | FFHQ | AFHQv2 | LSUN-bedroom | LSUN-cat |
|---|---|---|---|---|---|
| DDPM (Ho et al., 2020) | 1000 | 3.52 | - | 4.89 | 17.10 |
| ADM (Dhariwal & Nichol, 2021) | 1000 | - | - | 1.90 | 5.57 |
| NSCN++ (Song et al., 2021b) | 79 | 25.95 | 18.52 | - | - |
| DDPM++ (Song et al., 2021b) | 79 | 3.39 | 2.58 | - | - |
| EDM (Karras et al., 2022) | 79 | 2.39 | 1.96 | 3.57 | 6.69 |
| EDM (Karras et al., 2022) | 15 | 15.81 | 13.67 | - | - |
| Diff-Instruct (Luo et al., 2023c) | 1 | 19.93 | - | - | - |
| PD (Salimans & Ho, 2021) | 1 | - | - | 16.92 | 29.60 |
| CT (Song et al., 2023) | 1 | - | - | 16.00 | 20.70 |
| CD (Song et al., 2023) | 1 | 12.58 | 10.75 | 7.80 | 11.00 |
| **DisBack** | 1 | **10.88** | **9.97** | **6.99** | **10.30** |

In this stage, $G_{stu}$ and $s_\phi$ are also optimized alternately and the optimization of $s_\phi$ is the same as in the original score distillation (Eq. 5). Compared to existing score distillation methods, the final target of DisBack is the same while constraining the convergence trajectory to achieve more efficient distillation of the student generator. Algorithm 2 summarizes the distribution backtracking stage.

## 5 EXPERIMENT

Experiments are conducted on different models across various datasets. We first compare the performance of DisBack with other multi-step diffusion models and distillation methods (Sec. 5.1). Secondly, we compare the convergence speed of DisBack with its variants without the constraint of the convergence trajectory (Sec. 5.2). Thirdly, further experiments are conducted to demonstrate DisBack's effectiveness in mitigating the score mismatch issues (Sec. 5.3). Then, we also conduct the ablation study to show the effectiveness of introducing the convergence trajectory (Sec. 5.4). Finally, we show the results of DisBack on text-to-image generation tasks (Sec. 5.5).

### 5.1 QUANTITATIVE EVALUATION

DisBack can achieve performance comparable to or even better than the existing diffusion models or distillation methods. Experiments are conducted on different datasets. (1) The unconditional generation on FFHQ 64x64, AFHQv2 64x64, LSUN-bedroom 256x256 and LSUN cat 256x256. (2) The conditional generation on ImageNet 64x64. The performance of DisBack is shown in Tab. 1 and Tab. 2. All the DisBack models are distilled from the pre-trained EDM model (Karras et al., 2022).

For unconditional generation, the one-step generator distilled by the DisBack achieves comparable performance across different datasets compared to multi-step generation diffusion models. Specifically, it outperforms the original EDM model with 15 NFEs (10.88 of DisBack and

Table 2: The conditional generation performance of DisBack on ImageNet 64x64 dataset.

| Model | NFE ($\downarrow$) | FID ($\downarrow$) |
|---|---|---|
| DDPM (Ho et al., 2020) | 1000 | 3.77 |
| DDDM (Zhang et al., 2024) | 1000 | 2.11 |
| EDM (Karras et al., 2022) | 511 | 1.36 |
| EDM (Karras et al., 2022) | 15 | 10.46 |
| Moment Matching (Salimans et al., 2024) | 8 | 3.3 |
| SlimFlow (Zhu et al., 2024) | 1 | 12.34 |
| BOOT (Gu et al., 2024) | 1 | 12.30 |
| DDDM (Zhang et al., 2024) | 1 | 3.47 |
| CTM (Kim et al., 2024a) | 1 | 2.06 |
| Sid (Zhou et al., 2024) | 1 | 1.52 |
| DMD2 (Yin et al., 2024a) | 1 | 1.51 |
| Diff-Instruct (Luo et al., 2023c) | 1 | 5.57 |
| PD (Salimans & Ho, 2021) | 1 | 8.95 |
| CT (Song et al., 2023) | 1 | 13.00 |
| CD (Song et al., 2023) | 1 | 6.20 |
| **DisBack** | 1 | **1.38** |

15.81 of EDM on FFHQ64). Compared to existing one-step generators and distillation methods, DisBack achieves optimal performance. For conditional generation, the DisBack achieves the best

performance compared to the existing models. Moreover, DisBack requires no training data and additional constraints during training. In conclusion, DisBack can achieve competitive distillation performance compared to existing models.

## 5.2 CONVERGENCE SPEED

We conducted a series of experiments to demonstrate the advantages of DisBack in accelerating the convergence speed of the score distillation process on unconditional CIFAR10 (Krizhevsky, 2009), FFHQ 64x64 (Karras et al., 2019), and conditional ImageNet 64x64 (Deng et al., 2009) datasets. Diff-Instruct (Luo et al., 2023c) is the existing SOTA score distillation method, which can be regarded as a variation of DisBack not introducing the convergence trajectory. We compared the FID trends of DisBack and Diff-Instruct during the distillation process in the same situation.

The results are shown in Fig. 1. As for unconditional generation, DisBack achieves a convergence speed 1.97 times faster than the variant without the constraint of convergence trajectory on the FFHQ 64x64 dataset and 5.67 times faster on the CIFAR10 dataset. For the conditional generation on the ImageNet 64x64 dataset, DisBack is 1.79 times faster than the variant without the constraint of convergence trajectory. The fast convergence speed is because constraining the convergence trajectory of the generator provides a clear optimization direction, avoiding the generator falling into suboptimal solutions and enabling faster convergence to the target distribution.

## 5.3 EXPERIMENTS ON SCORE MISMATCH ISSUE

In this part, experiments are conducted to validate the positive impact of constraining the convergence trajectory on mitigating the mismatch issues. We propose a new metric called mismatch degree to assess whether the predicted score of the teacher model matches the distribution's real score given a data distribution. This score is inspired by the score-matching loss.

$$d_{mis} = \mathbb{E}_{\boldsymbol{x}_0 \sim q_0^G} \mathbb{E}_{\boldsymbol{x}_t \sim \mathcal{N}(\boldsymbol{x}_0, \sigma_t)} \| s_\theta(\boldsymbol{x}_t, t) - \nabla_{\boldsymbol{x}_t} \log q_t(\boldsymbol{x}_t) \|_2 \tag{9}$$

Here $\boldsymbol{x}_t$ is the noisy data from the assessed distribution. Besides, $s_\theta(\boldsymbol{x}_t, t)$ represents the predicted score and $\nabla_{\boldsymbol{x}_t} \log q_t(\boldsymbol{x}_t)$ represents the real score. As shown in Eq. (4) and (6), the closer $s_\theta(\boldsymbol{x}_t, t)$ is to $\nabla_{\boldsymbol{x}_t} \log q_t(\boldsymbol{x}_t)$, the more accurate the gradient estimation for the student generator obtains. Because the real scores are not available in practice, we use Stable Target Field (STF) (Xu et al., 2022) to approximate the real score $\nabla_{\boldsymbol{x}_t} \log p_t(\boldsymbol{x}_t)$ on the assessed distribution. STF estimation leverages reference batches to reduce the variance of training objectives, which has been proven to yield accurate asymptotically unbiased estimates of the real score (Xu et al., 2022).

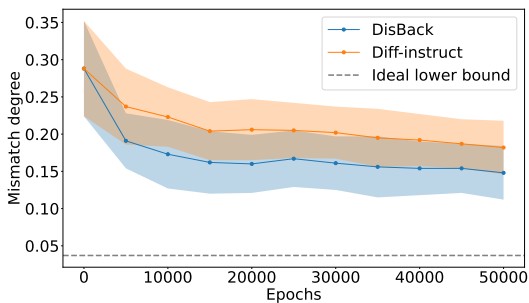

Figure 4: The mismatch degree during the distillation process of Diff-Instruct and proposed DisBack. The standard deviation is visualized. DisBack effectively mitigates the mismatch degree during the entire distillation process.

When the assessed distribution is close to the distribution of the teacher model $s_\theta$, the mismatch degree is small, and vice versa. When calculated directly on the training data, the resulting mismatch degree represents the ideal lower bound. Therefore, the mismatch degree can be used to assess the convergence degree of the generated distribution during the distillation process and visualize the convergence speed under the constraint of the convergence trajectory.

We conduct experiments on the FFHQ 64x64 dataset with Diff-Instruct (Luo et al., 2023c) as a baseline. We calculate the mismatch degree on the distribution of the student generator of both Diff-Instruct and the proposed DisBack. The pre-trained EDM model is chosen as the teacher model. In this scenario, the ideal lower bound of the mismatch degree is $0.037$. We visualized the mismatch degree in Fig. 4. With the constraining of the convergence trajectory, the mismatch degree of the proposed DisBack is lower during the distillation process, meaning the student generator con-

Table 3: Ablation study on constraining the convergence trajectory to the score distillation process. The FID ($\downarrow$) scores in each case are shown.

| Model | FFHQ | AHFQv2 | ImageNet | LSUN-bedroom | LSUN-cat |
|---|---|---|---|---|---|
| DisBack | **10.88** | **9.97** | **1.38** | **6.99** | **10.30** |
| w/o Convergence Trajectory | 12.26 | 10.29 | 5.96 | 7.43 | 10.63 |

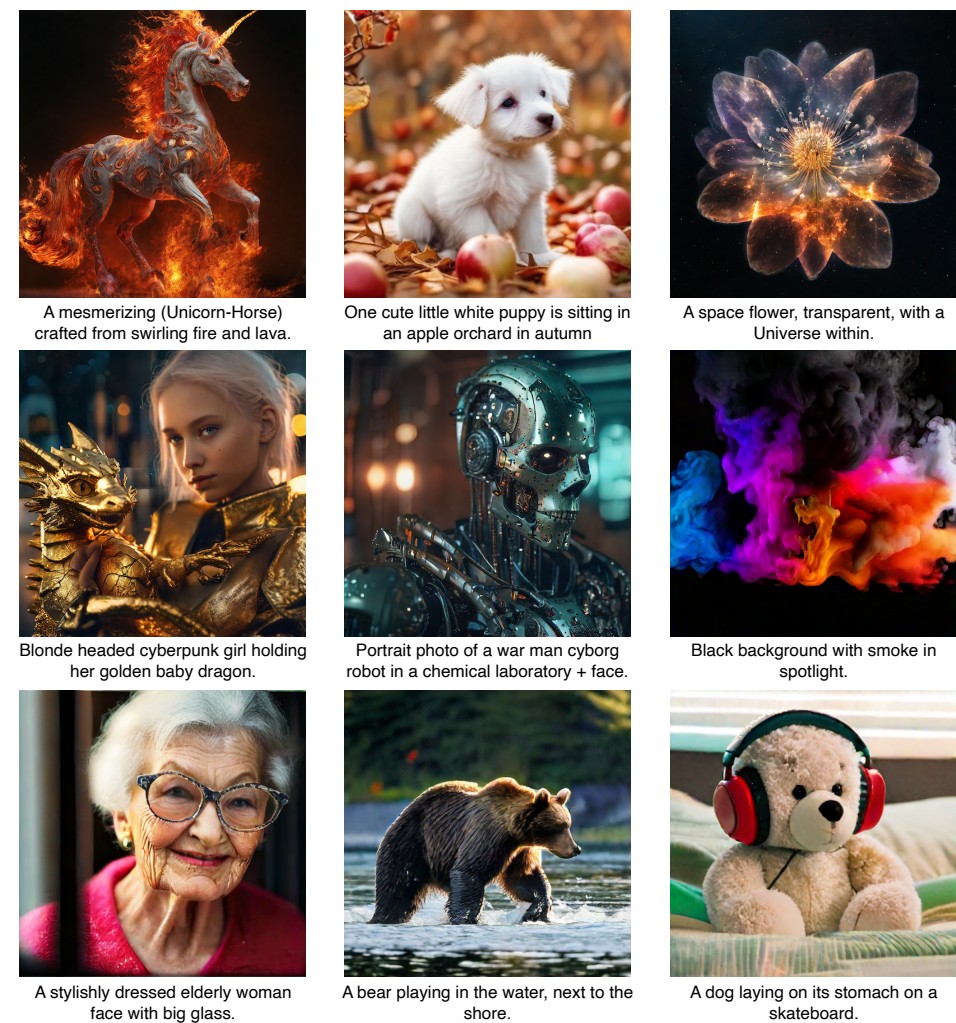

A mesmerizing (Unicorn-Horse) crafted from swirling fire and lava.

One cute little white puppy is sitting in an apple orchard in autumn

A space flower, transparent, with a Universe within.

Blonde headed cyberpunk girl holding her golden baby dragon.

Portrait photo of a war man cyborg robot in a chemical laboratory + face.

Black background with smoke in spotlight.

A stylishly dressed elderly woman face with big glass.

A bear playing in the water, next to the shore.

A dog laying on its stomach on a skateboard.

Figure 5: Generation samples by DisBack distilled from SDXL with 1024×1024 resolution.

verges faster and better. Thus, by constraining the convergence trajectory, the mismatch issue can be mitigated and DisBack can achieve more efficient distillation.

## 5.4 ABLATION STUDY

Ablation studies are conducted to compare the performance of DisBack with its variant without the constraint of the convergence trajectory. The results are shown in Tab. 3. Results show that the variant without the constraint of convergence trajectory suffers from a performance decay in different cases. This confirms the efficacy of constraining the convergence trajectory between the student generator and the teacher model can improve the final performance of the generation.

## 5.5 TEXT TO IMAGE GENERATION

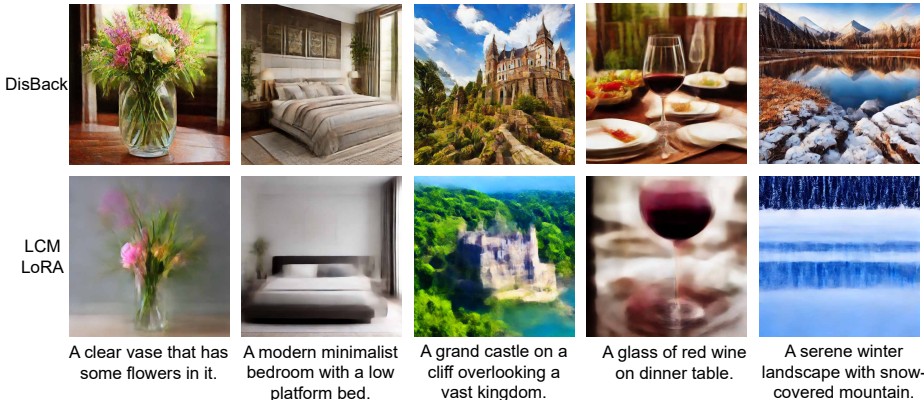

Figure 6: One step generation samples by original LCM-LoRA and its variant distilled from SD v1.5 with DisBack in 512×512. LCM-LoRA with DisBack can generate images with higher quality.

Further experiments are conducted on text-to-image generation tasks. We use DisBack to distill the SDXL model (Podell et al., 2024) and evaluate the FID scores of the distilled SDXL and the original SDXL on the COCO 2014 (Radford et al., 2021). The user studies are conducted to verify the effectiveness

Table 4: The results of text-to-image generation.

| Model | NFE (↓) | FID (↓) | Patch-FID (↓) | CLIP (↑) |
|---|---|---|---|---|
| LCM-SDXL Luo et al. (2023b) | 1 | 81.62 | 154.40 | 0.275 |
| LCM-SDXL Luo et al. (2023b) | 4 | 22.16 | 33.92 | 0.317 |
| DMD2 Yin et al. (2024a) | 1 | 19.01 | 26.98 | 0.336 |
| DMD2 Yin et al. (2024a) | 4 | 19.32 | **20.86** | 0.332 |
| SDXL-Turbo Sauer et al. (2025) | 1 | 24.57 | 23.94 | **0.337** |
| SDXL-Turbo Sauer et al. (2025) | 4 | 23.19 | 23.27 | 0.334 |
| SDXL Lightning Lin et al. (2024) | 1 | 23.92 | 31.65 | 0.316 |
| SDXL Lightning Lin et al. (2024) | 4 | 24.46 | 24.56 | 0.323 |
| SDXL Podell et al. (2024) | 100 | 19.36 | 21.38 | 0.332 |
| DisBack | 1 | **18.96** | 26.89 | 0.335 |

of DisBack. We randomly select 128 prompts from the LAION-Aesthetics (Schuhmann et al., 2022) to generate images and ask volunteer participants to choose the images they think are better. Detailed information about the user study is included in Sec. B.3. The results of the FID evaluation and user study are presented in Tab. 4. DisBack achieved optimal FID and comparable CLIP scores in single-step generation compared to baselines, but the Patch-FID showed a slight decay. The preference scores of DisBack over the original SDXL are 61.3%. Some generation samples are shown in Fig. 2 and Fig. 5.

We also conducted experiments on LCM-LoRA (Luo et al., 2023b). The LCM-LoRA distilled from SDv1.5 using DisBack has an FID score of 36.37 on one-step generation, while the FID score of the original LCM-LoRA is 78.26. Some generated samples of DisBack and original LCM-LoRA are shown in Fig. 6. The details of experiments and results are provided in Sec. A.1.

## 6 CONCLUSION

**Summary.** This paper proposes Distribution Backtracking Distillation (DisBack) to introduce the entire convergence trajectory of the teacher model in the score distillation. The DisBack can also be used to distill large-scale text-to-image models. DisBack performs a faster and more efficient distillation and achieves a comparable or better performance in one-step generation compared to existing multi-step generation diffusion models and one-step diffusion distillation models.

**Limitation.** The performance of DisBack is inherently limited by the teacher model. The better the original performance of the teacher model, the better the performance of DisBack will also be. Additionally, to achieve optimal performances in both accelerated distillation and generation quality, DisBack requires careful design of the distribution degradation path and the setting of various hyperparameters (such as how many epochs are used to fit each intermediate node in distribution backtracking stage). While with no meticulous design, it can also achieve better performance, further exploration is required to enable the model to reach optimal performance.

## 7 ACKNOWLEDGMENTS

This work is supported by the Provincial Key Research and Development Plan of Zhejiang Province under No. 2024C01250(SD2), the National Key R&D Program of China under No. 2022YFB3303301, and the National Natural Science Foundation of China under No. 62006208. This work is supported by Alibaba-Zhejiang University Joint Research Institute of Frontier Technologies. The authors would like to thank anonymous reviewers for their helpful comments.

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

Table 5: The performance of DisBack on the distillation from pre-trained EDM model to FastGAN on FFHQ, AFHQv2, and CelebA in the resolution of $64 \times 64$.

| Model | NFE ($\downarrow$) | FFHQ | | AFHQv2 | | CelebA | |
|---|---|---|---|---|---|---|---|
| | | FID ($\downarrow$) | IS ($\uparrow$) | FID ($\downarrow$) | IS ($\uparrow$) | FID ($\downarrow$) | IS ($\uparrow$) |
| FastGAN (Liu et al., 2020) | 1 | 30.27 | 2.37 | 28.59 | 5.94 | 29.35 | 2.36 |
| EDM (NFE 11) (Karras et al., 2022) | 11 | 29.28 | 2.97 | **13.67** | **10.86** | 23.05 | 3.01 |
| Score GAN (Franceschi et al., 2023) | 1 | 43.89 | 2.13 | 53.86 | 2.13 | 50.41 | 2.12 |
| **DisBack** | 1 | **23.84** | **3.27** | 18.95 | 7.00 | **23.16** | **3.02** |

# A  MORE EXPERIMENT RESULTS

## A.1  EXPERIMENT RESULTS ON TEXT-TO-IMAGE GENERATION

We conducted the experiment on LCM-LoRA (Luo et al., 2023b). LCM-LoRA is a Low-Rank Adaptation (LoRA) version of the Latent Consistency Model (LCM) Luo et al. (2023a), applicable across fine-tuned Stable Diffusion models for high-quality, single-step or few-step generation. In this experiment, we use LCM-LoRA as the student generator and Stable Diffusion v1.5 as the teacher model. We observed that the score distillation underperforms when LCM-LoRA serves as the teacher model. This issue likely stems from the infeasibility of directly converting the outputs of LCM-LoRA into scores.

We distill the LCM-LoRA with the proposed DisBack and evaluate the FID scores on the COCO 2014 dataset (Radford et al., 2021) with the resolution of 512×512. 50,000 real images and 30,000 generated images were used to calculate FID scores. The 30,000 generated images were obtained by generating one image for each of the 30,000 distinct prompts. In the case of one-step generation, the original LCM-LoRA has an FID score of 78.26, while the DisBack achieves an FID of 36.37. The change in FID scores over training steps is illustrated in Fig. 7, showing that DisBack achieves a 1.5 times acceleration in convergence speed and yields superior generation performance within the same training period.

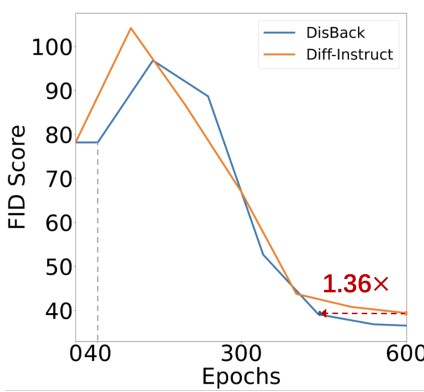

Figure 7: The FID scores of LCM-LoRA distilled from SD1.5 across training steps. The first 40 epochs refer to the computational overhead of the degradation recording stage of DisBack. DisBack achieves faster convergence and better performance.

## A.2  ADDITIONAL EXPERIMENT RESULTS

To explore the distillation performance of the proposed method when the architectures of the teacher and student models differ, we opt for EDM Karras et al. (2022) as the pre-trained diffusion model, and FastGAN Liu et al. (2020) architecture for the student model to conduct the experiment. Table 5 shows the performance of baselines and the proposed models on the distillation task from a diffusion model to a generator. Compared to the original FastGAN, DisBack can effectively improve the generation quality. The results also show that the one-step sampling performance of DisBack is better than Score GAN and EDM with 11 NFEs.

## A.3  VISUALIZATION OF INTERMEDIATE TEACHER TRAJECTORY

To further demonstrate the effectiveness of the degradation path, we visualized the images generated by the initial generator, the intermediate checkpoints and the teacher model, along the degradation path. As shown in Fig. 8 and 9, We can observe that the images generated by the first node in the trajectory are similar to those of the initial generator, while the images generated by the last node in the trajectory are close to those of the teacher model. This is consistent with our theoretical analysis.

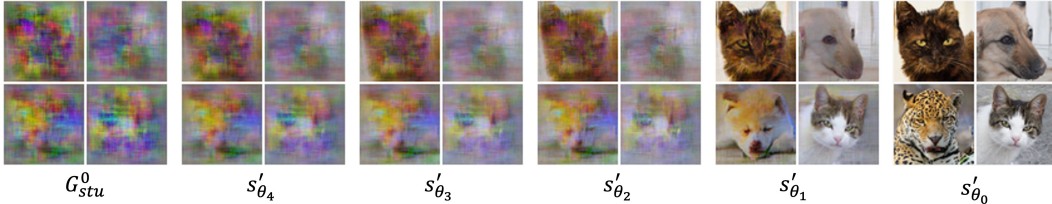

$G^0_{stu}$ $\quad$ $s'_{\theta_4}$ $\quad$ $s'_{\theta_3}$ $\quad$ $s'_{\theta_2}$ $\quad$ $s'_{\theta_1}$ $\quad$ $s'_{\theta_0}$

Figure 8: Samples from the initial generator, intermediate teacher trajectory nodes. Here $s'_{\theta_0}$ is the teacher model. The teacher model is the pre-trained EDM model on the AFHQv2 dataset, the student generator is FastGAN.

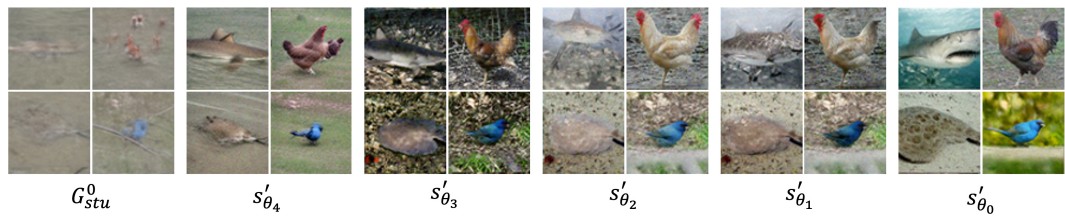

$G^0_{stu}$ $\quad$ $s'_{\theta_4}$ $\quad$ $s'_{\theta_3}$ $\quad$ $s'_{\theta_2}$ $\quad$ $s'_{\theta_1}$ $\quad$ $s'_{\theta_0}$

Figure 9: Samples from the initial generator, intermediate teacher trajectory nodes. Here $s'_{\theta_0}$ is the teacher model. The teacher model and the student generator are both the pre-trained EDM model on the ImageNet dataset.

## B  IMPLEMENTATION DETAILS

### B.1  DATASET SETUP

We experiment on the following datasets:

The FFHQ (Flickr-Faces-HQ) dataset (Karras et al., 2019) is a high-resolution dataset of human face images used for face generation tasks. It includes high-definition face images of various ages, genders, skin tones, and expressions from the Flickr platform. This dataset is commonly employed to train large-scale generative models. In this paper, we utilize a derivative dataset of the FFHQ called FFHQ64, which involves downsampling the images from the original FFHQ dataset to a resolution of 64×64.

The AFHQv2 (Animal Faces-HQ) dataset (Choi et al., 2020) comprises 15,000 high-definition animal face images with a resolution of 512×512, including 5,000 images each for cats, dogs, and wild animals. AFHQv2 is commonly employed in tasks such as image-to-image translation and image generation. Similar to the FFHQ dataset, we downscale the original AFHQv2 dataset to a resolution of 64×64 for the experiment.

The ImageNet dataset (Deng et al., 2009) was established as a large-scale image dataset to facilitate the development of computer vision technologies. This dataset comprises over 14,197,122 images spanning more than 20,000 categories, indexed by 21,841 Synsets. In this paper, we use the ImageNet64 dataset, a subsampled version of the ImageNet dataset. The Imagenet64 dataset consists of a vast collection of images with a resolution of 64×64, containing 1,281,167 training samples, 50,000 testing samples, and 1,000 labels.

The LSUN (Large Scale Scene Understanding) dataset (Yu et al., 2015) is a large-scale dataset for scene understanding in visual tasks within deep learning. Encompassing numerous indoor scene images, it spans various scenes and perspectives. The LSUN dataset comprises multiple sub-datasets, in this study, we use the LSUN Cat and Bedroom sub-datasets with a resolution of 256×256.

## B.2 EXPERIMENT SETUP

For experiments on FFHQ 64x64, AFHQv2 64x64, and ImageNet 64x64 datasets, the pre-trained models are provided by the official release of EDM Karras et al. (2022). We use Adam optimizers to train the student generator $G$ and $s_\phi$, with both learning rates set to $1e^{-5}$. The training consisted of 50,000 iterations on four NVIDIA 3090 GPUs, and the batch size per GPU is set to 8. The training ratio between $s_\phi$ and $G$ remains at $1 : 1$. In the Degradation stage, we trained for 200 epochs total, saving a checkpoint every 50 epochs, resulting in a total of 5 intermediate nodes along the degradation path $\{s'_{\theta_i} | i = 0, 1, 2, 3, 4\}$. In the Distribution Backtracking stage, when $i \geq 3$, each checkpoint was trained for 1,000 steps. When $i < 3$, each checkpoint was trained for 10,000 steps. The remaining steps were used to distill the original teacher model $s'_{\theta_0}$.

For experiments on LSUN bedroom and LSUN cat datasets, the pre-trained EDM models are provided by the official release of Consistency Model Song et al. (2023). During the training, we set $\sigma_{max}$ to 80 and keep it constant during the single-step generation process. We use SGD and AdamW optimizers during training to train the generator $G$ and $s_\phi$, with learning rates set to $1e^{-3}$ and $1e^{-4}$, respectively. The training consisted of 10,000 iterations on one NVIDIA A100 GPU, and the batch size per GPU is set to 2. The training ratio between $s_\phi$ and $G$ remains at $4 : 1$. In the Degradation stage, we trained for 200 epochs total and saved the checkpoint every 50 epochs, resulting in a total of 5 intermediate nodes along the degradation path $\{s'_{\theta_i} | i = 0, 1, 2, 3, 4\}$. In the Distribution Backtracking stage, when $i \geq 3$, each checkpoint was trained for 500 steps. When $i < 3$, each checkpoint was trained for 1000 steps. The remaining steps were used to distill the original teacher model $s'_{\theta_0}$.

When distilling the SDXL model, the teacher model and the student generator are both initialed by the pre-trained SDXL model on the huggingface (model id is 'stabilityai/stable-diffusion-xl-base-1.0'). We use Adam optimizers to train $G$ and $s_\phi$, with learning rates set to $1e^{-3}$ and $1e^{-2}$, respectively. The training consisted of 50,000 iterations on one NVIDIA A100 GPU, and the batch size per GPU is set to 1. The training ratio between $s_\phi$ and $G$ remains at $1 : 1$. The training prompts are obtained from LAION-Aesthetics. In the Degradation stage, we trained for 1,000 epochs total and saved the checkpoint every 100 epochs, resulting in a total of 10 intermediate nodes along the degradation path $\{s'_{\theta_i} | i = 0, 1, ..., 9\}$. In the Distribution Backtracking stage, each checkpoint was trained for 1,000 steps. The remaining steps were used to distill the original teacher model $s'_{\theta_0}$.

## B.3 USER STUDY SETUP

Firstly, we randomly selected 128 prompts from the LAION-Aesthetics (Schuhmann et al., 2022). Then we use the original SDXL model and the distilled SDXL model to generate 128 pairs of images. Subsequently, we randomly recruit 10 volunteers, instructing each to individually evaluate the fidelity, detail, and vividness of these pairwise images. 10 volunteers included 6 males and 4 females, aged between 24 and 29. 5 of them have artificial intelligence or related majors and the other 5 of them have other majors. They were given unlimited time for the experiment, and all of the volunteers completed the assessment with an average time of 30 minutes. Finally, we took the average of the evaluation results of 10 volunteers as the final user study result.

## C THEORETICAL DEMONSTRATION

### C.1 KL DIVERGENCE OF DISBACK

As the KL divergence follows

$$D_{\mathrm{KL}}(q \parallel p) = \mathbb{E}_q \left[ \log \frac{q}{p} \right] \tag{10}$$

The KL divergence of generated distribution and training distribution at timestep $t$ can be written as

$$
\begin{aligned}
D_{\mathrm{KL}} & \left( q_t^G \left( \boldsymbol{x}_t \right) \parallel q_t \left( \boldsymbol{x}_t \right) \right) \\
& = \mathbb{E}_{x_t \sim q_t^G(\boldsymbol{x}_t)} \log \frac{q_t^G \left( \boldsymbol{x}_t \right)}{q_t \left( \boldsymbol{x}_t \right)} \\
& = \mathbb{E}_{x_0 \sim G(z; \eta)} \left[ \log q_t^G \left( \boldsymbol{x}_t \right) - \log q_t \left( \boldsymbol{x}_t \right) \right] \\
& = \mathbb{E}_z \left[ \log q_t^G \left( \boldsymbol{x}_t \right) - \log q_t \left( \boldsymbol{x}_t \right) \right]
\end{aligned}
\tag{11}
$$

Thus, the gradient of KL divergence can be estimated as

$$\nabla_\eta D_{\text{KL}}\left(q_t^G\left(\boldsymbol{x}_t\right)\|q_t\left(\boldsymbol{x}_t\right)\right) = \mathbb{E}_{t,\boldsymbol{\epsilon}}\left[s_\phi\left(\boldsymbol{x}_t,t\right) - s_\theta\left(\boldsymbol{x}_t,t\right)\right]\frac{\delta\boldsymbol{x}_t}{\delta\eta} \tag{12}$$

## C.2 STABLE TARGET FIELD

Given $\boldsymbol{x}_0 \sim q_0$ is the training data, $\boldsymbol{x}_t \sim p(\boldsymbol{x}_t \mid \boldsymbol{x}_0)$ is the disturbed data, Xu *et al.* (Xu et al., 2022) presents an estimation of the score as:

$$\nabla_{\boldsymbol{x}_t}\log p_t(\boldsymbol{x}_t) = \frac{\nabla_{\boldsymbol{x}_t}p_t(\boldsymbol{x}_t)}{p_t(\boldsymbol{x}_t)} = \frac{\mathbb{E}_{\boldsymbol{x}_0}\nabla_{\boldsymbol{x}_t}p(\boldsymbol{x}_t \mid \boldsymbol{x}_0)}{p_t(\boldsymbol{x}_t)} \tag{13}$$

The transition kernel $p(\boldsymbol{x}_t \mid \boldsymbol{x}_0)$ follows the Gaussian distribution $p(\boldsymbol{x}_t \mid \boldsymbol{x}_0) \sim \mathcal{N}(\mu_t, \sigma_t^2 I)$. Here $\mu_t = \boldsymbol{x}_0$ in Variance Exploding SDE (Song et al., 2021b) but is defined differently in other diffusion models.

$$p(\boldsymbol{x}_t \mid \boldsymbol{x}_0) = \frac{1}{\sqrt{(2\pi^k)}\sigma_t}\exp(-\frac{(\boldsymbol{x}_t - \mu_t)^T(\boldsymbol{x}_t - \mu_t)}{2\sigma_t^2}) \tag{14}$$

$$
\begin{aligned}
&\nabla_{\boldsymbol{x}_t}p(\boldsymbol{x}_t \mid \boldsymbol{x}_0)\\
&= \nabla_{\boldsymbol{x}_t}\left[\frac{1}{\sqrt{(2\pi^k)}\sigma_t}\exp(-\frac{(\boldsymbol{x}_t - \mu_t)^T(\boldsymbol{x}_t - \mu_t))}{2\sigma_t^2}\right]\\
&= p(\boldsymbol{x}_t \mid \boldsymbol{x}_0)\nabla_{\boldsymbol{x}_t}(-\frac{(\boldsymbol{x}_t - \mu_t)^T(\boldsymbol{x}_t - \mu_t)}{2\sigma_t^2})\\
&= p(\boldsymbol{x}_t \mid \boldsymbol{x}_0)\frac{\mu_t - \boldsymbol{x}_t}{\sigma_t^2}
\end{aligned}
\tag{15}
$$

Combine Eq. (13) to Eq. (15), we have

$$\nabla_{\boldsymbol{x}_t}\log p_t(\boldsymbol{x}_t) = \mathbb{E}_{\boldsymbol{x}_0}\frac{p(\boldsymbol{x}_t \mid \boldsymbol{x}_0)}{p_t(\boldsymbol{x}_t)}\frac{\mu_t - \boldsymbol{x}_t}{\sigma_t^2} = \frac{1}{p_t(\boldsymbol{x}_t)}\mathbb{E}_{\boldsymbol{x}_0}p(\boldsymbol{x}_t \mid \boldsymbol{x}_0)\frac{\mu_t - \boldsymbol{x}_t}{\sigma_t^2} \tag{16}$$

Let $B$ be a set of reference samples for Monte Carlo estimation, we have

$$p_t(\boldsymbol{x}_t) = \mathbb{E}_{\boldsymbol{x}_0}p(\boldsymbol{x}_t \mid \boldsymbol{x}_0) \approx \frac{1}{|B|}\sum_{x_0^{(i)}\in B}p(\boldsymbol{x}_t \mid \boldsymbol{x}_0^{(i)}) \tag{17}$$

Combine the Eq. (16) and Eq. (17), we can get

$$\nabla_{\boldsymbol{x}_t}\log p_t(x_t) = \mathbb{E}_{\boldsymbol{x}_0}\frac{p(\boldsymbol{x}_t \mid \boldsymbol{x}_0)}{p_t(\boldsymbol{x}_t)}\frac{\mu_t - \boldsymbol{x}_t}{\sigma_t^2} \approx \frac{1}{p_t(\boldsymbol{x}_t)}\frac{1}{|B|}\sum_{x_0^{(i)}\in B}p(\boldsymbol{x}_t \mid \boldsymbol{x}_0^{(i)})\frac{\mu_t - \boldsymbol{x}_t}{\sigma_t^2} \tag{18}$$

Here the "$\approx$" represents the Monte Carlo estimate.

Depending on the network prediction, the diffusion model can be divided into different types, including $\epsilon$ prediction (Karras et al., 2022) and $x_0$ prediction (Song et al., 2021a; Ho et al., 2020; Nichol & Dhariwal, 2021). When the score $\nabla_{\boldsymbol{x}_t}\log p_t(x_t)$ is estimated by Eq.(18), it can be converted to $\epsilon$, $x_0$ and $v$ by a series of transformations.

$$\hat{\boldsymbol{\epsilon}} \approx -\sigma_t\nabla_{\boldsymbol{x}_t}\log p_t(x_t) \tag{19}$$

$$\hat{\boldsymbol{x}}_0 \approx \nabla_{\boldsymbol{x}_t}\log p_t(x_t) * \sigma_t^2 + \boldsymbol{x}_t \tag{20}$$

## D DISCUSSION

### D.1 THE GRADIENT ORIENTATION OF SCORE DISTILLATION

To better illustrate the process of training a generator with score distillation, we provide a more intuitive explanation. As shown in Figure 10, given a noisy data $\boldsymbol{x}_t$, the direction of $s_\phi(\boldsymbol{x}_t, t)$ points to the generated distribution and the direction of $s_\theta(\boldsymbol{x}_t, t)$ points to the training distribution. In Sec.E.3, we used 2D data to illustrate the gradient directions of $\boldsymbol{x}_t$ on the teacher model $s_\theta$ and the auxiliary diffusion model $s_\phi$. Thus, the direction of $s_\theta(\boldsymbol{x}_t, t) - s_\phi(\boldsymbol{x}_t, t)$ is from the generated distribution towards the training distribution. Ideally, if the predicted score of $s_\phi$ and $s_\theta$ are both correct, this gradient leads $G$ to update in the right direction.

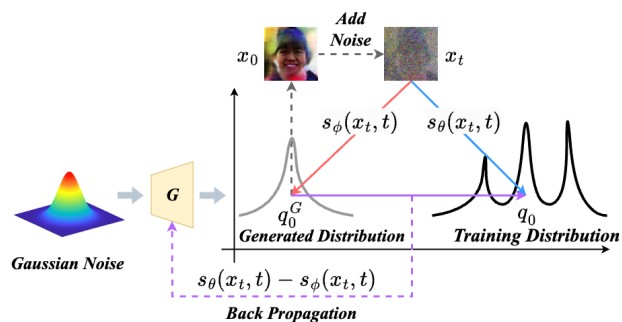

Figure 10: An illustration of the generator update by the minimization in Eq. (6).

## D.2 TWO TYPES OF MISMATCH ISSUE

We find that there are two types of mismatch issues during the training process: (1) The mismatch issue between the initially generated samples and $s_\phi$. (2) The mismatch issue between the initially generated samples and $s_\theta$. For the first type of mismatch issue, it prevents $s_\phi$ from accurately describing the distribution of the generator, leading to unreliable network predictions and introducing biases when computing the generator's gradients. The combined effect of these two mismatch issues exacerbates the negative impact on the optimization of the generator. Fortunately, among the three initialization methods of $s_\phi$ proposed in Section 3, when $s_\phi$ is initialized based on the generated images, its distribution becomes close to the initially generated distribution, which resolves the first type of mismatch issue. In this paper, we directly use the degraded teacher model $s'_{\theta_N}$ to initialize $s_\phi$. Because the degraded teacher model's distribution is closer to the initial generator's distribution, $s_\phi$ initialized by $s'_{\theta_N}$ can approximate the initial generation distribution and accurately predict the scores of the generated samples from the beginning. However, since $s_\theta$ remains fixed during the training, the second type of mismatch issue cannot be avoided through initialization. Consequently, in this paper, we mainly focus on discussing the second type of mismatch issue.

## D.3 TRAINING EFFICIENCY OF DISBACK

While DisBack involves an iterative optimization process during training, the optimization objective of $s_\phi(\boldsymbol{x}_t, t)$ aims to minimize the loss of the standard diffusion model based on Eq.(21), and the objective of student generator aims to minimize the KL divergence in Eq.(22). These two optimization processes do not entail adversarial training as in GANs. Consequently, the optimization process tends to be more stable. A recent work Monoflow (Yi et al., 2023) also discusses in GANs training a vector field is obtained to guide the optimization of the generator, but the vector field derives from the discriminator and the instability is not mitigated.

$$\min_\phi \mathbb{E}_{t,\boldsymbol{\epsilon}} \left\| s_\phi(\boldsymbol{x}_t, t) - \frac{\boldsymbol{x}_0 - \boldsymbol{x}_t}{\sigma_t^2} \right\|_2^2 \tag{21}$$

$$\min_\eta \mathbb{E}_{t,\boldsymbol{\epsilon}} D_{KL}\left(q_t^G(\boldsymbol{x}_t) \| q_t(\boldsymbol{x}_t)\right) \tag{22}$$

For the DisBack, training the student generator only requires two U-Nets to perform inference and subtraction. Training $s_\phi$ only involves training a single U-Net, and gradients do not need to be back-propagated to $G_{stu}$. Therefore, these models can be naturally deployed to different devices, making computational resource requirements more distributed. This ease of distribution allows for joint training on computational devices with limited capacity. In contrast, for GANs and VAEs, which require gradient propagation between models (discriminator to generator, decoder to encoder), computational requirements are more centralized, necessitating the use of a single device or tools like DeepSpeed to manage the workload.

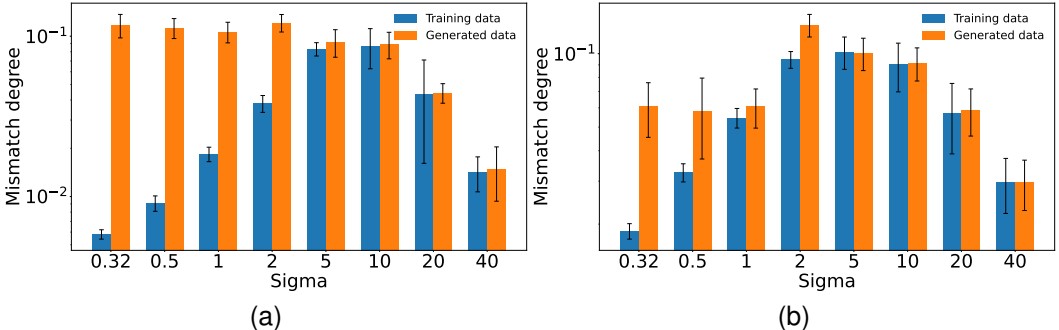

Figure 11: The results of the pre-experiments on mismatch degree. (a) $s_\theta$ and $G$ are both initialized by the pre-trained EDM (Karras et al., 2022) on FFHQ. (b) $s_\theta$ and $G$ are both initialized by the pre-trained EDM on ImageNet. The mismatch degree on the generated data is greater than on the training data, especially when the noise scale is low.

### D.4 VECTOR FIELD

In our research, each of the estimated score functions $s'_{\theta_i}$, for $i$ ranging from 0 to $N$, delineates a vector field $\mathbb{R}^{3 \times W \times H} \mapsto \mathbb{R}^{3 \times W \times H}$. We make a strong assumption behind our proposed method that these score functions represent existing or non-existent distributions and that they altogether imply a transformation path between $s_\theta$ and the student generator $G_{stu}^0$. Nevertheless, a score fundamentally constitutes a gradient field, signifying the gradient of the inherent probability density. A vector field is a gradient field when several conditions are satisfied, including path independence, continuous partial derivatives, and zero curls (Matthews, 1998). The vector field, as characterized by the score functions, may not meet these conditions, and thus there is not a potential function or a probability density function. Such deficiencies could potentially hinder the successful training of the student generator and introduce unforeseen difficulties in the distillation process. Specifically, in instances where $s_\theta$ does not precisely represent a gradient field, a highly probable scenario considering $s_\theta$ is a neural network, the samples generated from $s_\theta$ could encompass failure cases. Although our empirical studies exemplify the effectiveness of the proposed DisBack, the detrimental effects of the discussed issue remain unclear. We will further explore this issue in our future work.

### E ADDITIONAL DETAILS IN PRE-EXPERIMENTS

#### E.1 DISTRIBUTION MISMATCH ISSUES

Before conducting our research, we first carried out preliminary experiments to demonstrate that the proposed mismatch issue does indeed exist when using the endpoints of pre-trained diffusion models as teacher models. Using the method proposed in Eq. 9, we conducted experiments with the pre-trained EDM model on the ImageNet and FFHQ datasets. We calculated the mismatch degree separately on the student model's initial generated data and the teacher model's original training data, and the results are shown in Fig. 11. We can see that, on both datasets, the mismatch degree on the generated data of the pre-trained model is greater than that on the real training data, especially when the noise scale is small. This aligns with our hypothesis stated in Sec. 1, demonstrating that directly using the endpoint of a pre-trained model as the teacher model leads to a distribution mismatch problem and causes the unreliable predictions of the teacher model.

#### E.2 A TOY EXPERIMENT ON GAUSSIAN MIXTURE DISTRIBUTION

To validate the feasibility of the proposed DisBack, we conduct experiments on two-dimensional Gaussian mixture data. First, we randomly select 10 Gaussian distributions mixed as the training distribution $q_0$. Next, we construct a ResNet MLP as the two-dimensional diffusion model $s_\theta$ and train it using the created mixture Gaussian distribution. Similarly, we construct a simple MLP as the student generator $G_{stu}$ and train a model $s_\phi$ with the same architecture as $s_\theta$ using generated data.

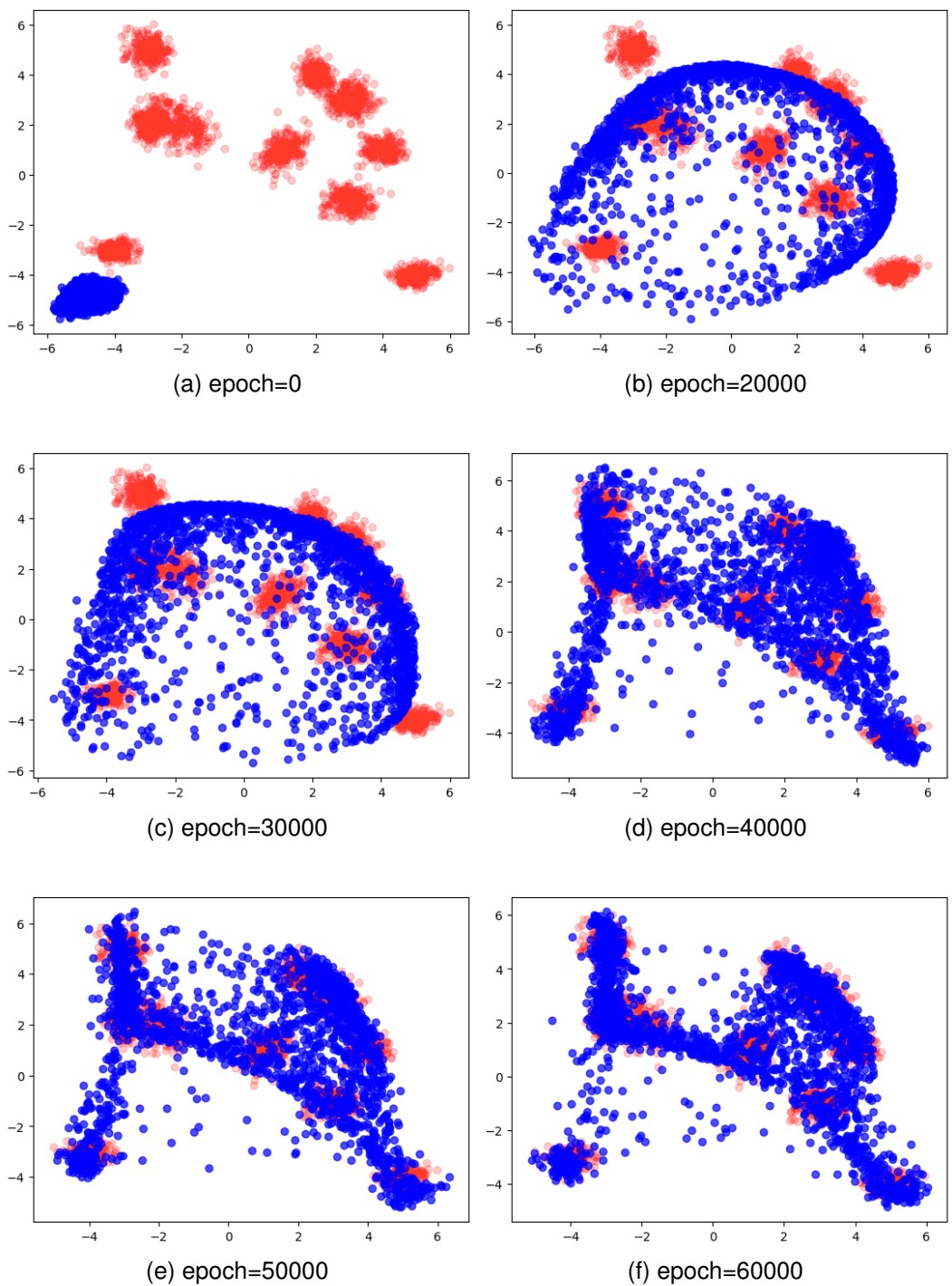

Figure 12: The distribution of student generator during the training process. Blue points visualize the generated distribution $q_t^G$ and the red points visualize the training distribution $q_0$.

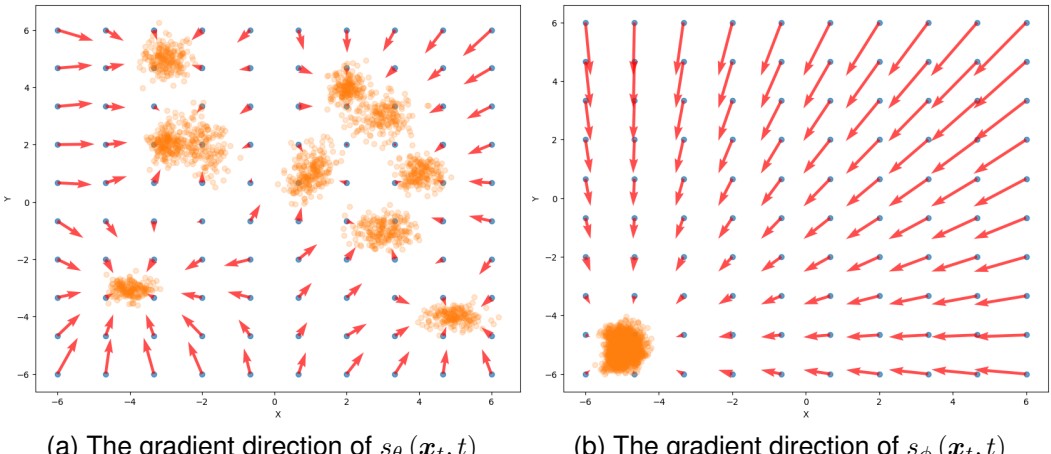

(a) The gradient direction of $s_\theta\left(\boldsymbol{x}_t, t\right)$      (b) The gradient direction of $s_\phi\left(\boldsymbol{x}_t, t\right)$

Figure 13: The gradient direction of $s_\theta\left(\boldsymbol{x}_t, t\right)$ and $s_\phi\left(\boldsymbol{x}_t, t\right)$ on $x_t$. The points in (a) are sampled from the training distribution and the points in (b) are sampled from the generated distribution.

Therefore, we can use $s_\theta$ and $s_\phi$ to train the student generator $G_{stu}$. During the training process, we visualize the distribution of the student generator and training data to intuitively demonstrate the changes in the student generator distribution under the proposed training framework. The distribution of $G_{stu}$ during the training process is shown in Figure 12. As training progresses, the generated distribution $q^G$ initially expands outward and then gradually convergents towards the training distribution. The results show that the proposed method for training the student generator is effective.

### E.3    GRADIENT ORIENTATION VERIFICATION OF DISBACK

As mentioned in Sec. 3, when updating $G_{stu}$ using Eq.(6), $s_\theta\left(\boldsymbol{x}_t, t\right)$ provides a gradient towards the training distribution, while $s_\phi\left(\boldsymbol{x}_t, t\right)$ provides a gradient toward the generated distribution.

To validate the correctness of these gradient directions, we experiment on two-dimensional data. We evenly sample $N$ data points within the range of $(x, y) \in [-6, 6]$ as the noisy data $\boldsymbol{x}_t$. Subsequently, we depict the gradient directions of $\boldsymbol{x}_t$ based on $s_\theta$ and $s_\phi$ respectively. As shown in Figure 13, consistent with theoretical derivation, for any given $\boldsymbol{x}_t$, the gradient direction of $s_\theta\left(\boldsymbol{x}_t, t\right)$ points toward the training distribution, and the magnitude of the gradient decreases as the distance to the training distribution decreases. Similarly, for any given $\boldsymbol{x}_t$, the gradient direction of $s_\phi\left(\boldsymbol{x}_t, t\right)$ points toward the generated distribution.

## F    ADDITIONAL SAMPLES FROM DISBACK

We provide additional samples from DisBack on FFHQ $64 \times 64$ (Figure 15), AFHQv2 $64 \times 64$ (Figure 16), ImageNet $64 \times 64$ (Figure 17), LSUN Bedroom $256 \times 256$ (Figure 18) and LSUN Cat $256 \times 256$ (Figure 19).

## G    ADDITIONAL ABLATION STUDY

We further explored the impact of different numbers of checkpoints along the degradation path and various degradation strategies on the performance of DisBack on the ImageNet64 dataset. The results are shown in Tab.6. When degrading for $200 iterations$ under default settings, the FID score with $N = 3$ intermediate checkpoints was 5.23, which outperformed the original Diff-instruct's FID of 5.57. The optimal FID of 1.38 was achieved with $N = 5$.

However, with $N = 11$, the FID score dropped to 8.35. When there are too many checkpoints in the path, the distributions of the checkpoints (e.g., $s'_{\theta_{10}}, s'_{\theta_{19}}, s'_{\theta_8}$) are very close to the distribution

Table 6: The conditional generation performance of DisBack on ImageNet 64x64 dataset.

| DisBack | FID ($\downarrow$) |
|---|---|
| N=1 | 5.96 |
| N=3 | 4.88 |
| N=5 | 1.38 |
| N=11 | 9.15 |
| N=11(Degradation iteration =400) | 244.72 |

of the initial generator. Training with these checkpoints provides limited progress for the generator. Furthermore, having too many checkpoints complicates the checkpoint transition scheduler during distillation, making it difficult to manage effectively. This often leads to inefficient updates for the generator across many iterations, wasting time without meaningful improvement. When the degradation iteration count is set to 400, DisBack fails to correctly distill the student generator. This is because excessive degradation iterations result in checkpoints near the initial generator on the degradation path being unable to generate sensible samples. Using these checkpoints for distillation provides no reasonable or effective guidance to the generator, ultimately causing it to fail to learn any meaningful information. In summary, when obtaining the degradation path, it is essential not to overly degrade the teacher model. The degraded teacher model's distribution should be close to the initial generator's distribution while still being able to produce relatively reasonable samples. Additionally, having too many or too few checkpoints along the path can adversely affect the final performance.

## H  FAILURE EXAMPLES

Fig. 14 presents several failure cases of DisBack.

In terms of FFHQ, AFHQv2, and ImageNet, while these images already capture the features of the corresponding datasets, the generated results lack accurate and clear backgrounds. The potential reasons for this include the fact that these datasets primarily focus on learning foreground content, with low requirements for image backgrounds, making the model difficult to clear backgrounds.

As for LSUN Cat and Bedroom, DisBack successfully generates details such as the cat's fur and the bed's texture, but it does not generate the overall shape and the detailed structure. This may be because the model does not capture the overall information of the data, only capturing local content. This issue may stem from the inherent limitations of U-Net, resulting in poor generation of overall structures in rare cases.

In the future, attempts will be made to use more advanced teacher models or improve the distillation algorithm to overcome these limitations. Moreover, we will further explore more advanced generator architectures such as StyleGAN Karras et al. (2020; 2021) to achieve higher-quality generation.

## I  ETHICAL STATEMENT

### I.1  ETHICAL IMPACT

The potential ethical impact of our work is about fairness. As "human face" is included as a kind of generated image, our method can be used in face generation tasks. Human-related datasets may have data bias related to fairness issues, such as the bias to gender or skin color. Such bias can be captured by the generative model in the training.

### I.2  NOTIFICATION TO HUMAN SUBJECTS

In our user study, we present the notification to subjects to inform the collection and use of data before the experiments.

> Dear volunteers, we would like to thank you for supporting our study. We propose the Distribution Backtracking Distillation, which introduces the convergence tra-

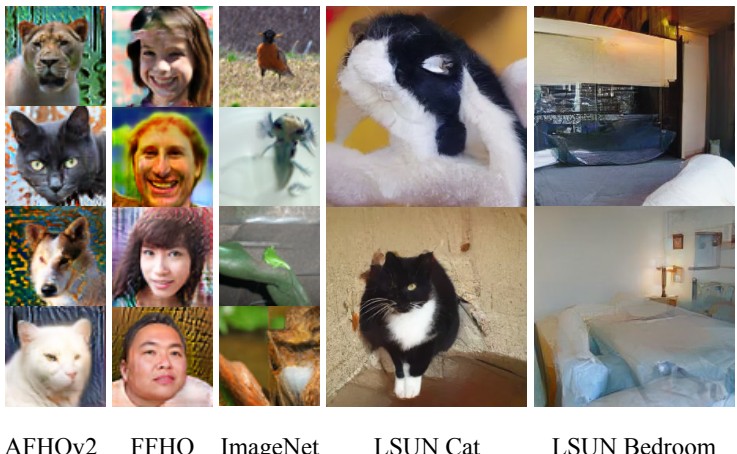

AFHQv2    FFHQ   ImageNet     LSUN Cat     LSUN Bedroom

Figure 14: Failure examples.

jectory into the score distillation process to achieve efficient and fast distillation and high-quality single-step generation.

All information about your participation in the study will appear in the study record. All information will be processed and stored according to the local law and policy on privacy. Your name will not appear in the final report. Only an individual number assigned to you is mentioned when referring to the data you provided.

We respect your decision whether you want to be a volunteer for the study. If you decide to participate in the study, you can sign this informed consent form.

The Institutional Review Board approved the use of users' data of the main authors' affiliation.

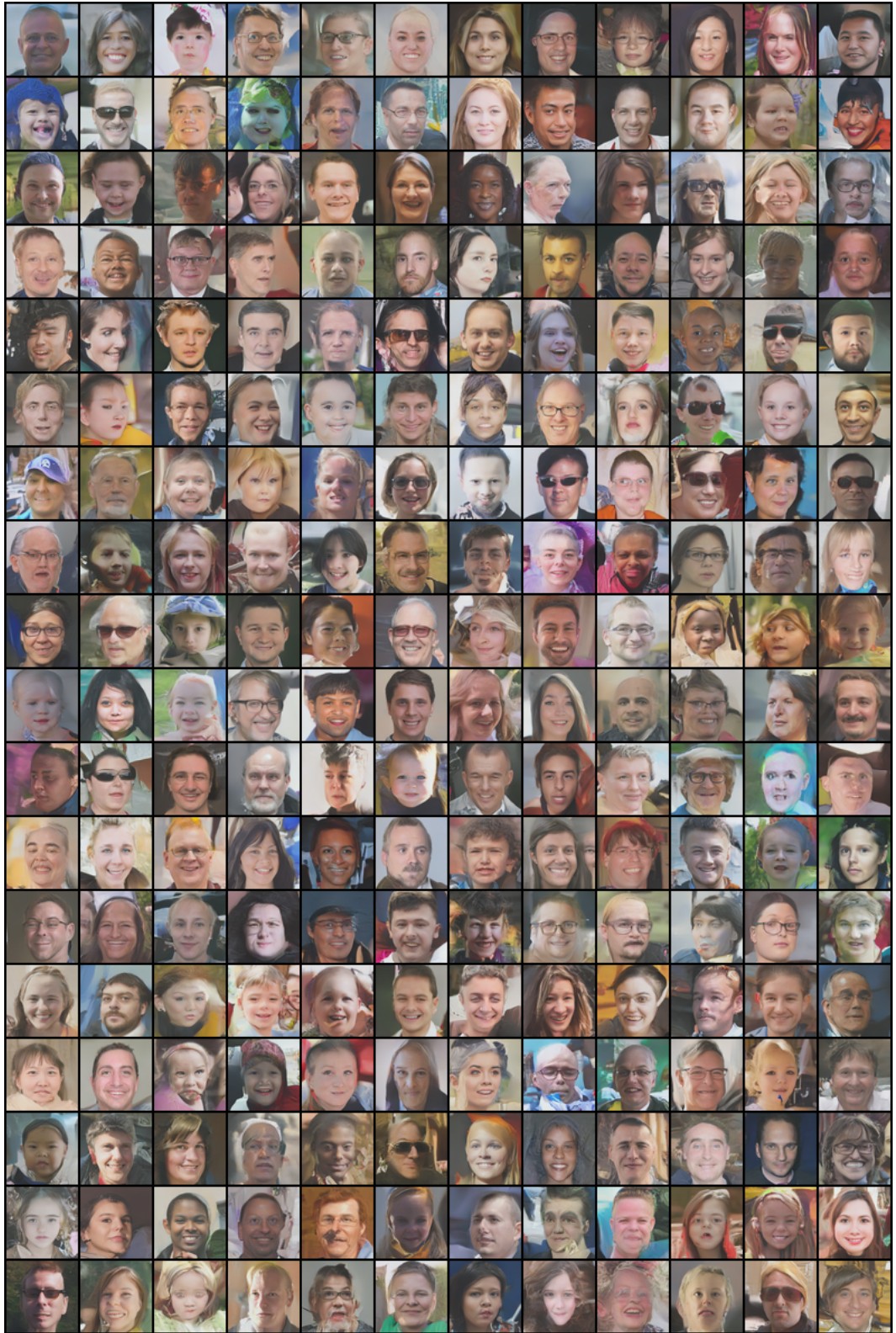

Figure 15: Additional Samples form conditional FFHQ 64x64.

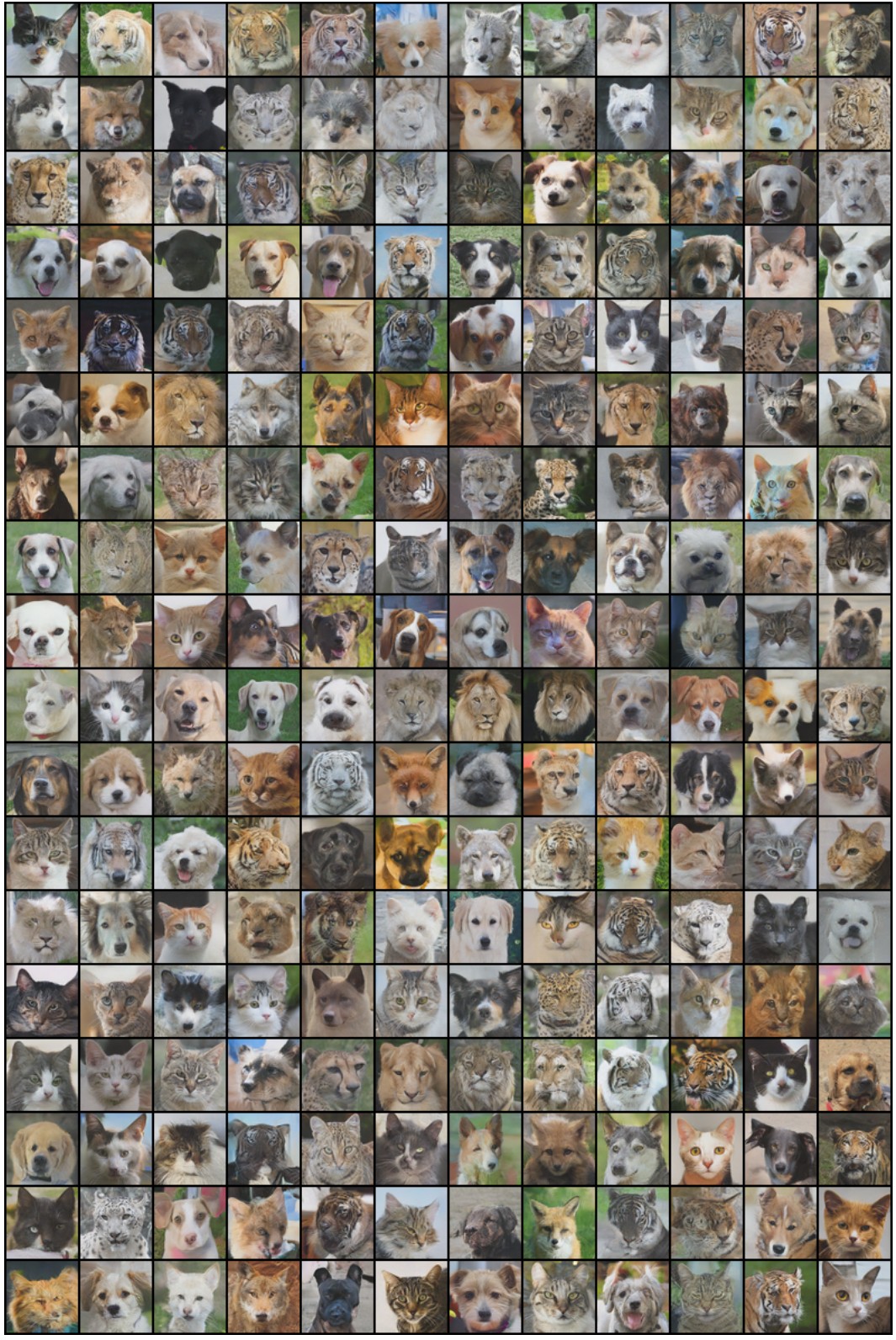

Figure 16: Additional Samples form conditional AFHQv2 64x64.

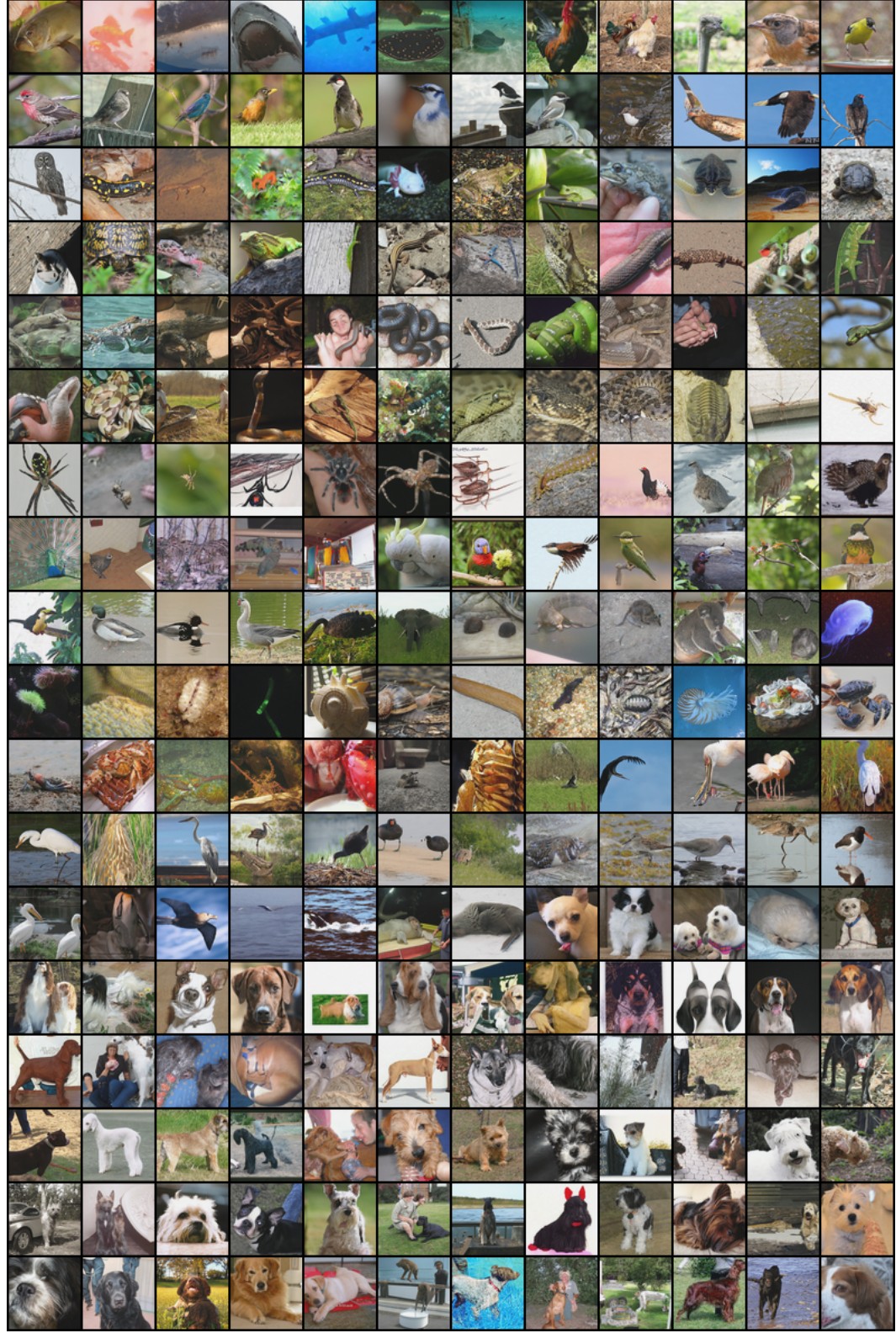

Figure 17: Additional Samples form conditional ImageNet 64x64.

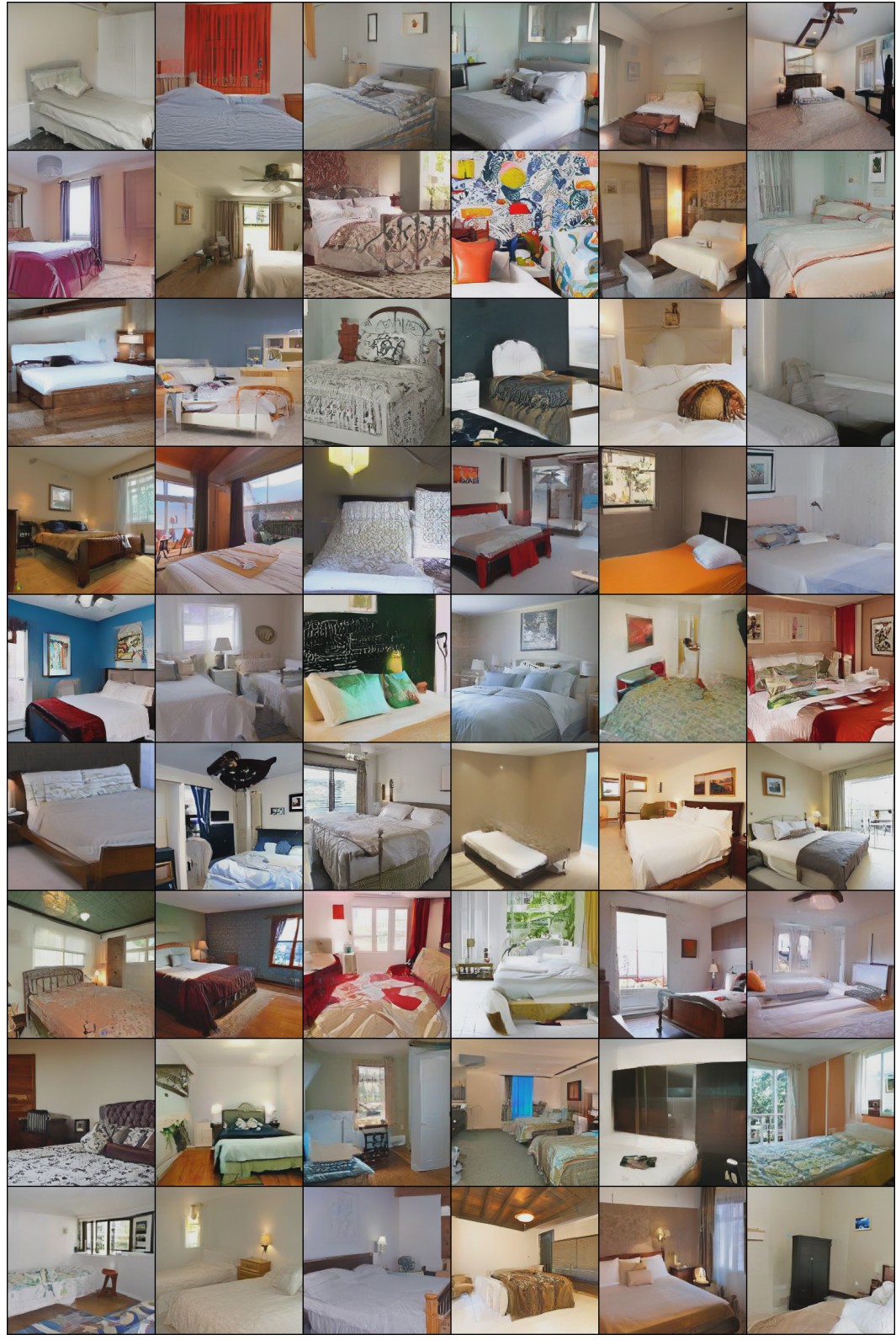

Figure 18: Additional Samples form conditional LSUN bedroom.

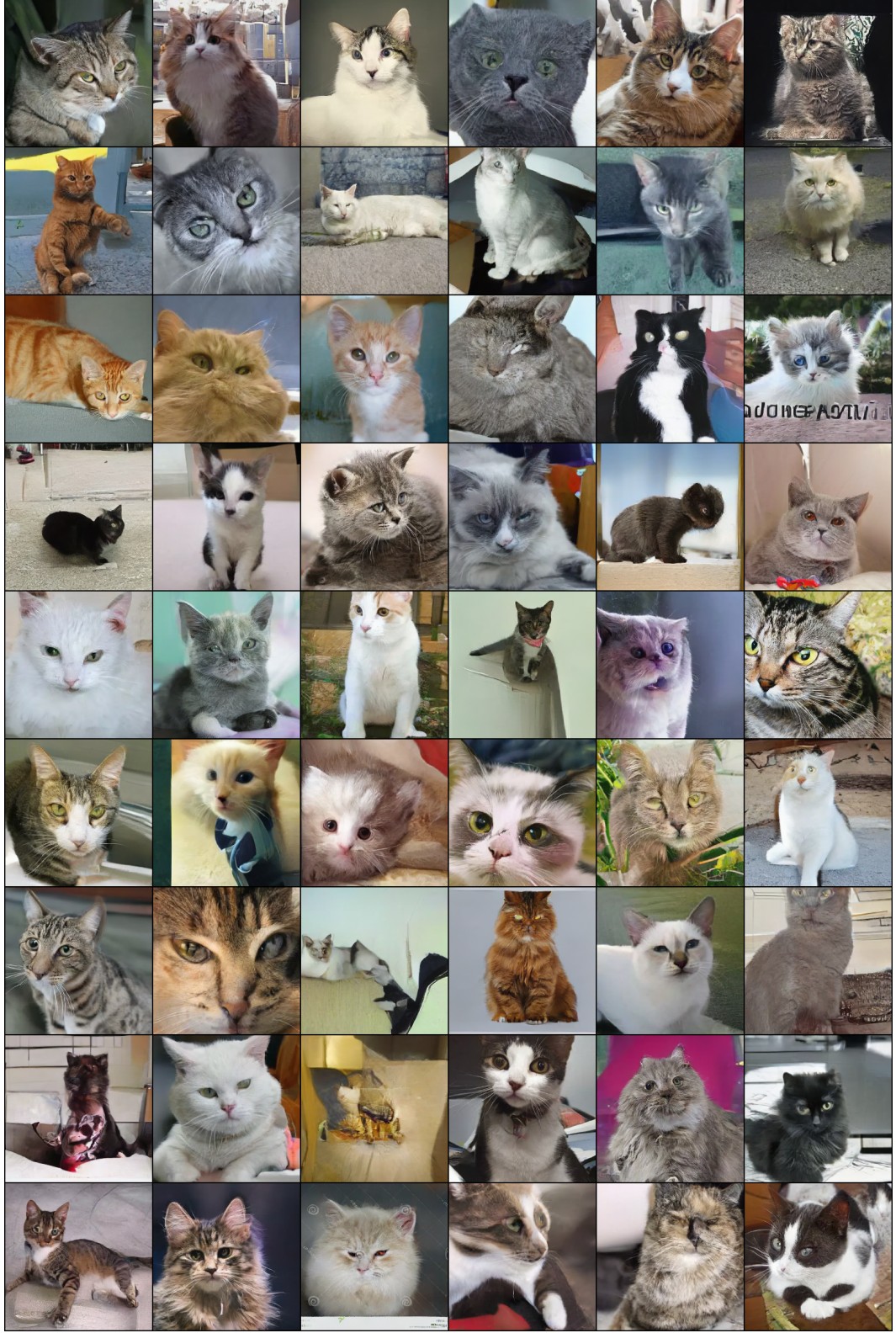

Figure 19: Additional Samples form conditional LSUN cat.

