# OpenReview forum: "Distribution Backtracking Builds A Faster Convergence Trajectory for Diffusion Distillation"
_ICLR.cc/2025/Conference — ICLR 2025 Poster_

### Official Review · Reviewer_HUTP · 2024-11-02

**Soundness:** 2
**Presentation:** 3
**Contribution:** 2
**Rating:** 5
**Confidence:** 3

**Summary:**

This paper introduces Distribution Backtracking Distillation (DisBack), a method to accelerate sampling in diffusion models by addressing the “score mismatch” issue common in traditional score distillation approaches. Unlike existing methods that rely solely on the endpoint of a pre-trained teacher model, DisBack captures the full convergence path between the teacher and student models. It does this through two stages: Degradation Recording, which records a degradation path from the teacher to the untrained student model, and Distribution Backtracking, where the student generator retraces this path to improve alignment with the teacher model.

**Strengths:**

1. DisBack addresses the common “score mismatch” issue in score distillation by incorporating the entire convergence trajectory. DisBack enables the student generator to align more accurately with the teacher model, leading to faster convergence and better optimization paths.
2. DisBack is designed to be easily integrated into current distillation frameworks, providing a versatile tool to further boost performance in generative model distillation.

**Weaknesses:**

**Major**:
- While the authors claim DisBack is orthogonal to those of other distillation methods, there is no evidence to support this point. It would be valuable if the authors could provide further experiments to show it can be incorporated into other distillation methods, like consistency distillation or adversarial score distillation.
- The paper aims to mitigate the score mismatch issue by employing degradation recording as convergence trajectory for distillation. The mismatch between the predicted score of generated samples and the model's prediction will be degraded but the mismatch between the model's prediction and the teacher's score prediction will be larger in degradation path.  This suggests a potential tradeoff between these two types of mismatches, which could impact the final model’s performance. Providing further analysis or empirical results on this point would strengthen the motivation and effectiveness of this approach.

**Minor**:
- In Eq.(6), $\partial x_t/\partial \eta$ should be included in expectation, same as (8).
- Better to use bold $\epsilon$ for noise and show the relationship between $\epsilon$ and $x_t$.
- In Algorithm 1&2, since the loss includes the expectation w.r.t. $t$ and $\epsilon$, the line to calculate $x_t = x_0 + \sigma_t \epsilon$ is unnecessary and misleading.
- Labels in Fig.7 are wrong.

**Questions:**

- The score estimation (7) is not general for all noising methods. For example, the score estimation of ddpm has a mean scale $\alpha_t$. When do distillation, should the teacher and student noising methods keep consistent?
- Compared with Diff-Instruct which only training student models to fit one teacher model, Algorithm 2 needs to fit $N-1$ intermediate checkpoints, what about the training overhead of this part? In Fig.1, did the epochs for DisBack in x-axis record from the degradation recording stage or from which other points?
- Any experiments to show the influence of the number of degradation checkpoints and the number of degradation epochs? Will more checkpoints and epochs mitigate the mismatch better?

---

> ### Author Response · Authors · 2024-11-19
>
> Thank you for your constructive review and valuable suggestions! Below, we provide detailed responses to your questions and comments.
>
> **[W1] DisBack is orthogonal to those of other distillation methods**
>
> **Response to W1**
>
> By “orthogonal to those of other distillation methods”, we mean that the proposed DisBack can be combined with existing score-distillation-based methods. We combined DisBack with ScoreGAN (Franceschi et al., 2024), distilling a pre-trained EDM model into a FastGAN generator, with the results presented in Appendix A.2. We conducted experiments on FFHQ, AFHQv2, and CelebA, achieving FID scores of 19.78, 18.95, and 20.55, respectively. The original FastGAN on these three datasets got 30.27, 28.59, and 29.35. We also compared our performance with EDM and ScoreGAN. Our DisBack also outperformed the EDM model with 11 NFE and surpassed the performance of ScoreGAN (Appndx A.2). This supports the orthogonality to other methods based on score distillation. We also conducted experiments on the Consistency Model (Song et al., 2023),. Due to time constraints, we distilled the ckpt of CD on ImgNet with DisBack. we achieved an FID of **5.73** for one-step generation, which outperforms the original Consistency Model’s FID of **6.20**. For Adversarial Score Distillation (ASD), it is a significant improvement of score distillation, with adversarial learning further introduced. Therefore, our proposed method can be applied to the score distillation component to mitigate the distribution mismatch issue. However, due to time constraints, we were unable to conduct further experiments on ASD.
>
> **[W2] A potential tradeoff between two types of mismatches.**
>
> **Response to W2**
>
> We argue that such a tradeoff does not exist. The target model changes from the degraded model back to the teacher model in the distillation. We give a brief formal explanation. In the degradation stage, we construct a series of diffusion models $\lbrace s_{\theta_i}' \mid i=0, \ldots, N\rbrace$, where $s_{\theta_0}^\prime = s_\theta$ approximates the teacher model's distribution $q_0$ and $s_{\theta_N}^\prime$ approximates the generated distribution $q^G_0$. Our discussed distribution mismatch issue is explained by $s_{\theta_0}^\prime \neq s_{\theta_N}^\prime$. Furthermore, notice that with a large $i$, $s_{\theta_i}'$ is closer to $s_{\theta_N}^\prime$, while with a small $i$, $s_{\theta_i}'$ is far away from $s_{\theta_N}^\prime$ but close to $s_{\theta_0}^\prime$. Therefore, given samples from $s_{\theta_N}^\prime$, as they are closer to $s_{\theta_{N-1}}^\prime$ than to $s_{\theta_0}^\prime$, the predicted scores given by $s_{\theta_{N-1}}^\prime$ are more accurate than that by $s_{\theta_0}^\prime$. As in Fig 3 and Sec 4.3, $G_{stu}^0$ is trained to fit $s_{\theta_i}'$ squentially for $i=N, \ldots, 0$ by switching between them. The 2nd type of "mismatch between the model's prediction and the teacher's score prediction" as commented by the reviewer is the difference between the intermediate ckpt $s_{\theta_i}'$ and the teacher $s_\theta$. This difference is not exposed to the student and is shrunk during the sequential switching in distillation.

---

> ### Author Response · Authors · 2024-11-19
>
> **[W3] Minor weaknesses.**
>
> **Response to W3**
>
> Thank you for your valuable suggestions. We have made the necessary revisions in the corresponding sections of the paper.​ For the labels in Figure 7, we used DisBack and Diff-instruct methods to distill LCM-LoRA and compared the FID scores. Therefore, the legend includes “DisBack” and “Diff-instruct.“​
>
>
> **[Q1] The general format of score estimation (7).**
>
> **Response to Q1**
>
> Here, we simply provide a formulation suitable for training predicted  $x_0$  models like EDM. Because the predicted score, $x_0$, and  $\epsilon$  can be converted into one another through linear transformations:
>
> $\text{score}_t = \frac{\alpha_t \hat{x}_0 - x_t}{\sigma_t^2}$
>
> $\text{score}_t = -\frac{\epsilon}{\sigma_t}$
>
> Thus, for methods like DDPM, which predict$\epsilon$, the degradation of the teacher model can be achieved simply by using the original DDPM training method
>
> $\mathcal{L}=\mathbb{E}| \epsilon- \epsilon_\theta (x_t, t) |^2$
>
> Our success on SDXL proves this point, as SDXL employs the predicted  $\epsilon$  approach, similar to DDPM. Additionally, since the student model can take any form (as we demonstrated in the experiment in Appendix A.2), the noise scheduler is determined by the teacher model.
>
> **[Q2] The training overhead of fitting N−1 intermediate checkpoints and the record starting point in Fig 1.**
>
> **Response to Q2**
>
> In the original Figure 1, we only recorded the epochs for Stage 2 and did not account for the overhead of Stage 1. However, as in Sec 4.2 model degradation follows the training of diffusion models, and our Stage 1 requires only 200 epochs. Even if the overhead of Stage 1 is included in Figure 1, our method still maintains a fast convergence speed. Based on your suggestion, we have updated Figure 1 to include the overhead from the degradation phase. Even with this adjustment, our method still converges faster than Diff-instruct. In Stage 2, the N-1 ckpts altogether only take a whole overhead of 50k iters, the same as Diff-Instruct. In practice, we have 5 ckpts. The target is switched at 1k, 2k, 12k, and 22k iters, and the final target is the teacher taking the rest 28k iters.
>
> **[Q3] The number of degradation checkpoints and the number of degradation epochs.**
>
> **Response to Q3**
>
> As mentioned in our paper, the number of degradation checkpoints may affect the final performance. We have highlighted this issue in the Limitation section. Empirically, setting too many or too few checkpoints can be detrimental. Too many checkpoints may cause the student model to spend excessive time on checkpoint transitions, leading to slower convergence. On the other hand, too few checkpoints may fail to address the distribution mismatch issue effectively. We are working diligently to complete this series of supplementary experiments, and the results will be presented during the rebuttal asap. For the number of degradation epochs, we find 200 iterations is enough for degradation to coverge.  Taking our suggested setting of these hyperparameters can improve distillation performance.

---

> > ### Comment · Reviewer_HUTP · 2024-11-25
> > **Official Comment by Reviewer HUTP**
> >
> > Thank you for your detailed reply.
> >
> > My concern has mostly been addressed. Regarding W1, please ensure meticulous consideration of your method’s applicability. Considering other reviewers’ comments on the experimental results, I will keep my scores unchanged.

---

> > > ### Author Response · Authors · 2024-11-25
> > >
> > > Thank you again for your valuable suggestions on our paper. In response to other reviewers’ comments on the experimental results, we have made supplementary adjustments and submitted a revised version. We kindly hope you to reconsider the score of our paper. If you have any questions about our responses or any other concerns regarding our paper, please feel free to comment and let us know. We will do our best to address your concerns promptly.

---

### Official Review · Reviewer_JLT9 · 2024-11-03

**Soundness:** 3
**Presentation:** 3
**Contribution:** 3
**Rating:** 3
**Confidence:** 5

**Summary:**

This paper propose a DisBack, which is a new distillation method of diffusion models. On the top of the Diff-Instruct, this DisBack propose a better training algorithm. While Diff-Instruct only use the pre-trained diffusion teacher, DisBack makes a series of degraded teachers, and use that teachers iteratively. This makes the student models easy to learn a teacher distribution.

**Strengths:**

- The idea is simple but effective. It is makes sense that distilled from degraded teacher make the distillation faster.
- Degradation Recording algorithm looks reasonable. The degraded teacher finally converges at the initialized student distribution, which make the student easy to learn in the early stage.
- The result compared to Diff-instruct seems the algorithm is effective.

**Weaknesses:**

- My major worry is that I can not trust the performance. The paper distilled from EDM in ImageNet 64 which have 2.44 FID, but the distilled student has the performance of 1.38 FID. In my understanding, I think the upper bound performance of the student is teacher.

- I also can not believe user preference study compared to the SDXL teacher. How can it better than teacher?

- The ablation on the number of degraded teacher N is missing. I want to see progressive performance boosting from N=1 (equivalent to the Diff-Instruct) to N= large.

- Is there any scheduling algorithm that changes the teacher in stage 2? It may requires lots of trials to find the schedule that determine when to change that target teacher from degraded to the original.

- Figure 1 is a little bit over-claimed. This algorithm should contains the training costs of stage 1.

**Questions:**

My major worry is the reliability of performance and the laborious algorithm. Please respond to my worries.

---

> ### Author Response · Authors · 2024-11-19
>
> We sincerely appreciate the comprehensive feedback provided by the reviewers. In response to constructive comments, we intend to resolve these issues by offering the following clarifications.
>
> **[W1] Student's performance is upperbounded by the teacher's.**
>
> **Response to W1**
>
> The FID 2.44 of EDM shown in Tab 2 is adapted from the Consistency Model (Song et al., 2023), not the original EDM paper. The official EDM achieves an FID of 1.36 on ImgNet64, which is lower than our 1.38. We adopt the official EDM ckpt as the teacher. Therefore, the performance of our student model does not surpass that of the teacher model. We have made the corrections of EDM performance in the revised version.
>
> **[W2] The user preference for the student model is better than the teacher model.**
>
> **Response to W2**
>
> There are existing works on score distillation where the student model outperforms the teacher model. For instance, in SwiftBrush [a], the authors used score distillation to distill the SD2.1 model into a single-step generator, ultimately achieving comparable or even higher scores in the Human Preference Score. Similarly, in the SiD [b], the authors distilled the generative capabilities of a pre-trained diffusion model into a single-step generator, ultimately surpassing the FID performance of the original teacher diffusion model. There are also other papers with similar conclusions [c-i]. The reason for this situation is that the student model inherits the capabilities of the teacher model, incorporating its advantages while discarding its shortcomings. Therefore, the student model may generate images that are more favored by users, resulting in an increase in user preference.
>
>  *a. Nguyen T H, Tran A. Swiftbrush: One-step text-to-image diffusion model with variational score distillation. CVPR2024*
>
>  *b. Zhou M, Zheng H, Wang Z, et al. Score identity distillation: Exponentially fast distillation of pretrained diffusion models for one-step generation. ICML2024.*
>
>   *c. Xie Q, Liao Z, Deng Z, et al. MLCM: Multistep Consistency Distillation of Latent Diffusion Model. arXiv:2406.05768, 2024.*
>
>   *d. Wang Z, Li Z, Mandlekar A, et al. One-Step Diffusion Policy: Fast Visuomotor Policies via Diffusion Distillation. arXiv:2410.21257, 2024.*
>
>   *e. Kim D, Lai C H, Liao W H, et al. Consistency trajectory models: Learning probability flow ode trajectory of diffusion. ICLR2024*
>
>  *f. Chang J, Wang S, Xu H M, et al. Detrdistill: A universal knowledge distillation framework for detr-families. CVPR2023*
>
>   *g. Salimans T, Mensink T, Heek J, et al. Multistep Distillation of Diffusion Models via Moment Matching. arXiv:2406.04103, 2024.*
>
>   *h. Dong J, Koniusz P, Chen J, et al. Adversarially Robust Distillation by Reducing the Student-Teacher Variance Gap. ECCV2025*
>
>   *i. Wang J, Chen Y, Zheng Z, et al. CrossKD: Cross-head knowledge distillation for object detection. CVPR2024*
>
> **[W3] The ablation study of the number of degraded teacher N.**
>
> **Response to W3**
>
> Thanks for your suggestion. We are working diligently to complete these experiments and will update them in the paper asap.
>
> **[W4] How to schedule intermediate ckpts along the convergence trajectory.**
>
> **Response to W4**
>
> W4：As in Appndx B.2, we only have 5 ckpts. In the 2nd stage, the first two ckpts are trained 1000 iters and the second two another 10,000 iters. The remaining iterations are taken by the original teacher. As we pointed out in the Limitation, the number of iterations each intermediate checkpoint is trained for and when to transition to the next checkpoint may affect performance. However, empirically we find the difference is small, and the proposed strategy is not sensitive to the scheduling. Although finding an optimal scheduler requires the search for a hyperparameter, adopting our default scheduler can still improve the performance of score distillation.
>
> **[W5] Containing the training costs of stage 1 in Figure 1.**
>
> **Response to W5**
>
> Thank you for your suggestion. As mentioned in Sec 4.2, our Stage 1 requires only 200 epochs. Even if these 200 epochs are included in the revised Fig 1, our method still converges faster than existing methods. We have revised Fig 1 to include the training time for Stage 1.

---

> > ### Comment · Reviewer_JLT9 · 2024-11-24
> >
> > Thank you for the response.
> >
> > **[W1]** OK. However, the NFE of 79 for the 1.36 performance EDM is likely incorrect. Please refer to the EDM paper and make the necessary corrections. If my memory serves me correctly, the NFE should be 511. If your performance of 1.38 is indeed accurate, I am willing to increase the score. However, I find it hard to trust a 0.02 FID gap within the distillation framework. In papers like CTM [e], they achieved better performance than the teacher by introducing a discriminator that follows the true data distribution, but this paper does not do that. Please demonstrate how the performance improves as N increases gradually from 1 to 5. Alternatively, show a curve in Fig 1-(c) where the performance approaches 1.38 as the epochs increase. That would make it more credible for me.
> >
> > **[W2]** OK.
> >
> > **[W3]** I really looking forward to see this.
> >
> > **[W4]** I also think that might be the case, but it would be good to show consistent performance across various scheduling scenarios. I'm somewhat skeptical of arguments that are only verbal.
> >
> > **[W5]** OK.
> >
> > # More Weakness
> >
> > - 6. The baselines in Table 4 seem insufficient. There are many baselines like SDXL-lightning, LCM-LoRA, Pixart-delta, DMD2, Distilling diffusion into conditional GANs, and SDXL-Turbo, whose checkpoints are available on Hugging Face. Additionally, the metrics also appear lacking. The FID at 1024 resolution is not a reliable metric because the inception network for FID supports an input of 299 px, leading to downsampling. This makes it impossible to reflect the high frequency of the samples. To overcome this, SDXL-Lightning proposed a metric called patch-FID, and I recommend measuring it. Furthermore, the text-image alignment score is missing. There are many alignment scores recently published, such as the outdated CLIP metric or more recent ones like compbench [1] or vqa scores. I strongly suggest adding these.
> >
> > - 7. I like the explanation of Figures 12 and 13. The matching process with an inaccurate teacher, which has a relatively wide support area, will help in matching the score vector field across the entire data space. If you connect this with the score mismatching mentioned in DLSM [2] or the score matching at biased data points mentioned in TIW-DSM [3] and write the related work, I think the motivation for the methodology will be more convincing.
> >
> > - 8. Will you release the code? Or can I see your code now?
> >
> > I hope the authors can address my concerns well so that I can be inclined to raise the score.
> >
> > [1][NIPS 2023] T2I-CompBench: A Comprehensive Benchmark for Open-world Compositional Text-to-image Generation
> >
> > [2][ICLR 2022] Denoising Likelihood Score Matching for Conditional Score-based Data Generation
> >
> > [3][ICLR 2024] Training unbiased diffusion models from biased dataset

---

> > > ### Author Response · Authors · 2024-11-25
> > >
> > > We would like to express our heartfelt thanks for your valuable comments. We provide a detailed explanation of your confusion below.
> > >
> > > **[W1.1] The NFE of EDM**
> > >
> > > **Response to W1.1**
> > >
> > > Thank you for pointing this out. The NFE of the 1.36 performance EDM is indeed 511, we have corrected this in the paper.
> > >
> > > **[W1.2] Why the distillation performance is improved without a discriminator**
> > >
> > > **Response to W1.2**
> > >
> > > Score distillation is equivalent to adversarial training with the objective of minimizing the KL divergence of GANs. Let h(·) be the discriminator, given a$x = G(z)$is the generated sample, the training objective of h(·) is
> > >
> > > $\min L = - E_x [\log h(x)] - E_{z \sim p_z}[\log (1-h(g(z)))]$
> > >
> > > Let $q_0$ and $q^G_0$ be the training distribution and the generated distribution, the optimal discriminator is
> > >
> > > $h(x) = \frac{q_0(x)}{q_0 (x)+q_0^G (x)}$
> > >
> > > the gradient of the generator $G$ is
> > >
> > > $\frac{\partial}{\partial \eta} L=E_z \nabla_{x}\left(\log \frac{1-h(x)}{h(x)}\right) \frac{\partial G(z)}{\partial \eta} $
> > >
> > > Because $\log \frac{1-h\left(x\right)}{h\left(x\right)} = \log \frac{q_0^G(x)}{q_0(x)}$, the gradient can be written as
> > >
> > > $\frac{\partial}{\partial \eta} L = E_z \nabla_{x}\left(\log \frac{q^G_0(x)}{q_0(x)}\right)\ \frac{\partial G(z)}{\partial \eta}$
> > >
> > > $=E_x\left[\nabla_{x} \log q_0^G(x)-\nabla_{x} \log q_0(x)\right] \frac{\partial G(z)}{\partial \eta}$
> > >
> > > $=E_{z,x}\left[s_\phi(x)-s_\theta(x)\right] \frac{\partial x}{\partial \eta}$
> > >
> > > Where $s_\theta(x)=\nabla_{x} \log q_0(x)$ and $s_\phi(x) = \nabla_{x} \log q_0^G(x)$ are the score function of the generator and the training distribution. From the gradient formulation of GAN generators, it is mathematically equivalent to the score distillation gradients of the student generator (as in Eq.(4) and Eq.(6) in our paper).
> > >
> > > $\nabla_\eta D_{KL} \left(q^G_{t} \left( x_t \right) \| q_{t}\left(x_t\right)\right) = E_{t,\epsilon} \left[\left[ \nabla_{x_t}\log q^G_{t}\left(x_t\right) - \nabla_{x_t} \log q_{t}\left(x_t\right)\right] \frac{\partial x_t}{\partial \eta}\right] \approx E_{t,\epsilon}\left[\left[ s_{\phi}\left(x_{t}, t\right)- s_{\theta}\left(x_{t}, t\right)\right] \frac{\partial x_{t}}{\partial \eta}\right]$
> > >
> > > In score distillation, $s_\phi$  can be seen as a specialized form of a discriminator, constraining the generator to produce samples whose scores closely match those of the pre-trained diffusion model. Additionally, score distillation overcomes issues such as mode-dropping that are commonly associated with GANs. When both $s_\theta$ and $s_\phi$ can accurately predict the scores of generated samples (This is precisely the goal of our paper), score distillation can achieve performance comparable to adversarial learning. Therefore, by using the DisBack method to mitigate the mismatch issue, our performance has significantly improved compared to Diff-instruct. A similar discussion is also present in Diff-instruct (Corollary 3.5 and Appendix A.4)

---

> > > ### Author Response · Authors · 2024-11-25
> > >
> > > **[W1.3] The ablation study of N**
> > >
> > > **Response to W1.3**
> > >
> > > We have completed the experiment on different $N$, the number of checkpoints on the degradation path. In our default setting, the DisBack degradation process involved 200 iterations, with $N=5$ checkpoints in the path (saving one checkpoint every 50 iterations). We evaluated the performance on the ImageNet64 dataset for $N=3$ (saving one checkpoint every 100 iterations) and  $N=11$  (saving one checkpoint every 20 iterations).  Due to time limitations, we were unable to conduct experiments for  $N=2$  and $N=4$. Additionally, we extended the degradation process to 400 iterations and evaluated the performance for  $N=11$ (saving one checkpoint every 40 iterations).  The results are presented in Appendix G of the revised paper.
> > >
> > > |N=1|N=3|N=5(default setting)|N=11|N=11(Degradation iteration=400)|
> > > |-|-|-|-|-|
> > > |5.96|4.88|1.38|9.15|244.72|
> > >
> > > For $N=3$, DisBack achieved an FID of 4.88, while for  $N=11$, the FID increased to 9.15.  The performance degradation at  $N=11$ results from that when there are too many checkpoints in the path, the distributions of certain checkpoints (e.g.,  $s_{\theta_{10}}^\prime, s_{\theta_{9}}^\prime, s_{\theta_{8}}^\prime$) are very close to the distribution of the initial generator.  Training with these checkpoints provides limited progress for the generator.  Furthermore, having too many checkpoints complicates the checkpoint transition scheduler during distillation, making it difficult to manage effectively.  This often results in inefficient updates to the generator across many iterations, wasting time without achieving meaningful improvements. When the degradation iteration count is set to 400, we observed that the student model could not be effectively trained.  This is because excessive degradation iterations lead to checkpoints near the initial generator on the degradation path being unable to generate reasonable samples.  Using these checkpoints for distillation fails to provide the generator with meaningful or effective guidance, ultimately resulting in the generator’s inability to move quickly towards the training distribution within limited iteration.
> > >
> > > Therefore, when obtaining the degradation path, it is crucial not to over-degrade the teacher model.
> > > We agree there is another balance concerned about the number of degradation iterations. If $s_{\theta_{N}}^\prime$ is too far away from $q_G^0$, our discussed mismatch issue arises. Simultaneous, if $s_{\theta_{N}}^\prime$ is too close to  $q^G_0$, $s_{\theta_{N}}^\prime$degrades too much and fails to give useful guidance to the student. In our extensive experiments on 5 datasets, we empirically find using our default setting (5 checkpoints in 200 degradation iterations) is likely to result in improved performance.
> > > We argue that to dillate along a convergence trajectory is not more difficult than to tune an extra adversarial loss with a new discriminator like CTM, because DisBack does not introduce any new loss or new component.

---

> > > ### Author Response · Authors · 2024-11-25
> > >
> > > **[W6] The baselines in Table 4**
> > >
> > > **Response to W6**
> > >
> > > Thank you for pointing this out, we are currently calculating the relevant Patch-FID and CLIP scores. Once the calculations are complete, we will update these results in the paper as soon as possible before the rebuttal deadline.
> > >
> > > **[W7] More explanation of the mismatch issue connecting DLSM and TIW-DSM**
> > >
> > > **Response to W7**
> > >
> > > Thank you for your suggestion. We have added a new section in the Related Work to discuss the score mismatch mentioned in DLSM and the score matching at biased data points mentioned in TIW-DSM, comparing these with the perspectives presented in our paper.
> > >
> > > DLSM is designed for conditional generation with classifier guidance. DLSM analyzes the score mismatch between the posterior score $\nabla_x \log p(x\mid y;\theta,\psi)$ and the estimated score with $\nabla_x \log p(y\mid x;\theta,\psi) + s_\theta(x)$ where $\psi$ is the parameter of the classifier.  To solve the issue,  Denoising Likelihood Score Matching loss is proposed to train the classifier for more accurate conditional score estimation. TIW-DSM is designed for training an unbiased diffusion model from the biased dataset. TIW-DSM highlights that dataset biases cause mismatches between the target score $\nabla_x \log p^t_{data} (x_t)$ and $\nabla_x \log p^t_{bias} (x_t)$. To mitigate this issue,  TIW-DSM introduces time-varying importance weights and score correction terms by assigning different weights to samples at each time step and correcting the scores. Our DisBack is designed for the score distillation of diffusion models. DisBack identifies the mismatch issue between the teacher model’s predicted scores $s_\theta(x_t,t)$  and the real scores $\nabla_x \log q^t(x_t)$ on generated samples. To solve this issue, DisBack introduces the convergence trajectory between the student generator and the pre-trained diffusion model. In summary, DLSM, TIW-DSM and our DisBack all find score mismatch issues on the targeted tasks and solve the issues with different strategies.
> > >
> > > **[W8] Releasing the code**
> > >
> > > **Response to W8**
> > >
> > > We have attached the code for training SDXL in the supplementary materials. Please download the zip file from OpenReview. This code was adapted from the DMD framework, and we distilled an SDXL model using it. The degradation recording stage of DisBack is in Ln 357-471 in main/degradation.py and the distribution backtracking is in Ln 322-622 in main/train_sd.py.
> > >
> > > With less than 3 days remaining until the rebuttal deadline, if you have any questions about our responses or any other concerns regarding our paper, please feel free to comment and let us know. We will do our best to address your concerns promptly.

---

> > > ### Author Response · Authors · 2024-11-29
> > > **Gentle reminder**
> > >
> > > Dear reviewer JLT9:
> > >
> > > We sincerely appreciate the time and effort you dedicated to reviewing our paper. In response to your concerns, we have provided additional experimental results and theoretical analysis for demonstrating the superiority of our framework.
> > >
> > > As the discussion period concludes in two days, we kindly request, if possible, that you review our rebuttal at your convenience. Should there be any further points requiring clarification or improvement, please know that we are fully committed to addressing them promptly. Thank you once again for your invaluable contribution to our research.
> > >
> > > Warm regards,
> > >
> > > The Authors

---

> ### Author Response · Authors · 2024-11-27
>
> **[W6] The baselines in Table 4**
>
> **Response to W6**
>
> We have updated the baseline and metric in the paper. The new table is as follows:
> |Model|NFE|FID|Patch-FID|CLIP|FAED|
> |-|-|-|-|-|-|
> |LCM-SDXL | 1 | 81.62 | 154.40 | 0.275|60.72|
> |LCM-SDX | 4 | 22.16 | 33.92 | 0.317|-|
> |DMD2 | 1 | 19.01 | 26.98 | 0.336|18.52|
> |DMD2 | 4 | 19.32 | 20.86 | 0.332|-|
> |SDXL-Turbo | 1 | 24.57 | 23.94 | 0.337|33.33|
> |SDXL-Turbo | 4 | 23.19 | 23.27 | 0.334|-|
> |SDXL Lightning | 1 | 23.92 | 31.65 | 0.316|36.20|
> |SDXL Lightning | 4 | 24.46 | 24.56 | 0.323|-|
> |SDXL | 100 | 19.36 | 21.38 | 0.332|-|
> |DisBack | 1 | 18.96 | 26.89 | 0.335|18.50|
>
> The baselines of LCM, SDXL-Turbo, SDXL Lightning and SDXL are referenced from DMD2. DisBack achieved optimal FID and comparable CLIP scores compared to existing models,  but the Patch-FID showed a slight decay.
>
> We also examine the models with Frechet Auto-Encoder Distance (FAED), a metric specifically designed for high-resolution images [1]. FAED encodes images into the latent space using a VAE and then computes the Fréchet distance. Compared to traditional FID, FAED does not downsample the images, ensuring that high-frequency details of the samples are preserved during evaluation. This allows FAED to reflect the overall quality of the holistic images more effectively. FAED can also be adaptively applied to evaluate multi-modal data that lacks a labeled dataset. Due to time constraints, we only obtain FAED scores of SDXL Turbo, SDXL Lightning, DMD2, and LCM under single-step sampling as baselines. Compared to other methods, DisBack achieved lower FAED score. This shows DisBack achieves better overall visual quality.
>
> [1] Oh C et al., Bips: Bi-modal indoor panorama synthesis via residual depth-aided adversarial learning, ECCV 2022

---

### Official Review · Reviewer_vTDX · 2024-11-03

**Soundness:** 2
**Presentation:** 2
**Contribution:** 2
**Rating:** 6
**Confidence:** 4

**Summary:**

In this paper, a novel approach is proposed to further improve existing diffusion distillation methods. The proposed approach leverages the convergence trajectory from teacher model to the initial state of student model to guide the training of student model backwards, which mitigates the score mismatching problem at the beginning of the distillation. Empirical results have demonstrated the superior performance of the proposed approach.

**Strengths:**

1. The paper has a clear motivation: to address the initial score mismatch problem in diffusion distillation. Specifically, the authors first identify the suboptimal performance of existing diffusion distillation methods as being due to the score mismatch issue at the beginning of student model training, and then propose a novel approach to resolve it.
2. The proposed approach is intuitive and effective. It makes sense to follow the degradation path from the teacher model to the student model in reverse during distillation, providing a progressive learning signal for the student and mitigating the initial score mismatch problem. In practice, by introducing this backtracking strategy, the distillation process is shown to be significantly faster than its variant without this technique.
3. The proposed approach is versatile, as it is orthogonal to other diffusion distillation approaches.

**Weaknesses:**

1. The state-of-the-art claim for the proposed method is misleading. According to the experimental setup in Appendix B.2, the proposed method trains for 500,000 **epochs** ($\approx500K \times \frac{|D|}{|B|}$ **iterations or steps**, where $|D|$ is the data size and $|B|$ is the batch size). This number is significantly higher than for the baselines. For example, Diff-Instruct only trains for ${\color{red}50K}$ **iterations** on ImageNet $64\times 64$, while DisBack (this paper) uses about ${\color{red}500K\times 40K=20G}$ **iterations** ($|D|=1,281,167$ and $|B|=32$), which is approximately $40,0000$ times larger. Even if "epochs" actually refers to steps (if it is a typo), it still represents 10 times the training length compared with the Diff-Instruct baseline. Additionally, the result (${\color{green}1.51}$) of DMD2 on ImageNet $64\times 64$ is achieved by training for ${\color{red}200K}$ **iterations**. With the extended training setup (${550K}$ **iterations** in total), DMD2 could achieve an FID of ${\color{green}1.28}$, which is lower than DisBack's ${\color{green}1.38}$. This raises concerns that the proposed strategy may not fully account for the state-of-the-art performance showcased. This is also supported by their ablation study and Table 3. The variant ("w/o convergence trajectory") is essentially Diff-Instruct, as noted in the main text on Lines 391-392. However, even this variant, when trained under the same setting, shows better performance on FFHQ (12.26) versus the original Diff-Instruct (19.93).

2. The speedup shown in Figure 1 is only plotted for epochs 0 to 2000, which covers only the early stage of the distillation. More epochs, either until convergence or until training budgets are exhausted, are needed to better understand how the backtracking strategy behaves throughout training.

3. Although the entire concept revolves around backtracking the degradation path, in actual training, each intermediate checkpoint is only trained for as few as $1000$ steps (for FFHQ, AFHQv2, and ImageNet at $64\times64$ resolution), while the remaining steps are trained with the original teacher model. This means that the proposed backtracking is used for only a small fraction of the student model's training, which makes it even harder to attribute the superior performance to the proposed strategy.

**Questions:**

See Weaknesses.

---

> ### Author Response · Authors · 2024-11-19
>
> Thank you for your constructive review and valuable suggestions! Below, we provide a detailed response to your questions and comments. If any of our responses fail to sufficiently address your concerns, please inform us, and we will promptly follow up.
>
> **[W1]  The number of training iterations.**
>
> **Response to W1**
>
> Thank you for pointing this out.  The number of epochs in Appndx B.2 should be **50k** instead of **500k**, which was a typo.  We have made corrections.  Our model is built on Diff-instruct, and we do NOT use a dataloader to read training data during distillation.  Each epoch contains only a single iteration.  Therefore, the total number of our training iterations is the same as that of Diff-Instruct as **50k**.  Moreover, the superior performance of DMD2 is due to its extra discriminator to constrain the student and 550k training iterations.  In contrast, our method focuses only on improving score distillation w/o additional modules or more iterations.  Our DisBack's 1.38 w/ 50k iters is lower than DMD2's 1.51 w/ 200k iters on ImgNet64, and DisBack's 12.26 is lower than Diff-Instruct's 19.93 both w/ 50k iters on FFHQ.  Our strategy fully accounts for our performance.
>
> **[W2] The range of training epochs shown in Figure 1.**
>
> **Response to W2**
>
> We included only the first 2000 steps because FID changes most significantly during this phase. In the rest of the training, the FID of DisBack and Diff-instruct drops gradually w/o significant fluctuation, but our DisBack converges to a lower FID finally. Our training consists of 50k iterations. If we plotted all of them, given the span of FID ranging from 1 to 170 in ImgNet64, the curve of DisBack and Diff-instruct would look too close in most areas due to space limitations. This is not conducive to visualizing the performance gain.
>
> **[W3] The number of iterations for intermediate checkpoints and the teacher model.**
>
> **Response to W3**
>
> Recall that our total training iterations is **50k**, not **500k**. As in Appndx B.2, we use 5 intermediate checkpoints during training. For checkpoints where i=3,4, we train **1,000** steps per checkpoint, but for checkpoints where i=1,2, we train for **10,000** steps per checkpoint. Therefore, the 4 checkpoints account for **22,000** iterations (taking up 44%), leaving only **28,000** iterations trained with the original teacher model. As in Sec 4.1, the mismatch issue is significant during the early stage of distillation. Thus during the first half of distillation, Disback effectively mitigates the issue, so the student converges much faster than Diff-Instruct as in Fig 1, gaining predominant advantages. Therefore, the superior performance is attributed to our proposed strategy.

---

> ### Comment · Reviewer_vTDX · 2024-11-24
>
> Thank you for your responses. While I appreciate the effort, my questions remain insufficiently addressed, and some points in your replies have deepened my confusion. Consequently, I have adjusted my rating to 3. However, I am open to reconsidering my rating based on the following clarifications:
>
> > Our model is built on Diff-Instruct... Therefore, the total number of our training iterations is the same as that of Diff-Instruct as **50k**.
> >
> > Our strategy fully accounts for our performance.
>
> If this is accurate, could you clarify why "DisBack w/o Convergence Trajectory" in **Table 3** achieves an FID of **12.26** on FFHQ, which is substantially better than the **19.93** FID reported for Diff-Instruct, both with 50k iterations?
>
> > DisBack's 12.26 is lower than Diff-Instruct's 19.93 both w/ 50k iters on FFHQ.
>
> In your response, you stated that DisBack achieves an FID of **12.26**, but Table 3 attributes this value to the "w/o Convergence Trajectory" variant, while DisBack itself achieves **10.88**. Could you confirm if this is another typo? Additionally, the improvement from 12.26 to 10.88 is less significant than the leap from 19.93 to 12.26, where the performance gain seems unrelated to the convergence trajectory.
>
> > ..., and we do NOT use a dataloader to read training data during distillation. Each epoch contains only a single iteration.
>
> Finally, based on the code in your supplementary material, it appears that you adopted the DMD2 codebase, which introduces distinct training strategies compared to Diff-Instruct, such as the two time-scale update rule and randomly sampled data labels. This makes it difficult to isolate the performance gains attributable to DisBack. For example, Diff-Instruct uses a dataloader to preserve the true label distribution, which is critical for datasets like ImageNet, where the label distribution is highly unbalanced. Could you clarify the rationale for using randomly sampled data labels instead? If there is any misunderstanding here, please point it out.
>
> To further clarify the unique advantages of your method, I request the training configuration files for experiments on FFHQ and ImageNet, which are missing from the supplementary material.

---

> > ### Author Response · Authors · 2024-11-25
> >
> > We sincerely thank you for your suggestions. Below, we provide additional explanations to address your concerns.
> >
> > **[W1] The FID score in Table 3**
> >
> > **Response to W1**
> >
> > We are sorry for this confusion. The FID of 19.93 in Table 1 is obtained by training the original [Diff-instruct implementation](https://github.com/pkulwj1994/diff_instruct/tree/main)  on the FFHQ dataset. The 12.26 of ``w/o convergence trajectory'' in Table 3 refers to the variant where the convergence trajectory is not used to distill the student generator in the second stage of distribution backtracking. However, the degradation path in the first stage was still preserved. In this case,  $s_\phi$ was initialized by the degraded teacher model $s_{\theta_N}'$ rather than the original teacher model $s_{\theta}$ as done by Diff-Instruct. In summary, the difference between 19.93 and 12.26 lies in the initialization of $s_\phi$.
> > The reason why the initialization of $s_\phi$ causes such a difference is another type of mismatch issue during the distillation process. In addition to DisBack addressing the mismatch between the initially generated samples and the teacher model, there is also a score mismatch issue between  $s_\phi$  and the generator.
> > In practice, $s_{\phi}$ has three initialization strategies: (1) $s_{\phi}$ is randomly initialized [1]. (2) $s_{\phi}$ is initialized as $s_{\theta}$ or its LoRA [2,3]. (3) $s_{\phi}$ is initialized by fitting the generated samples of the student generator, which is adopted in our paper.
> > If the teacher model or a random model is used to initialize  $s_\phi$,  $s_\phi$ will still produce inaccurate predicted scores on the initial generated samples, leading to suboptimal optimization of the student. whereas, when  $s_\phi$  is initialized using the degraded teacher model  $s_{\theta_N}'$,  $s_\phi$ approximates the initial generation distribution and accurately predicts the scores of the generated samples from the beginning. Score distillation is performed with both $s_{\theta}$ and $s_\phi$, and the alleviated mismatch issues of both score prediction networks lead to boosted performance. This explains why the FID in the “w/o convergence trajectory” setting is better than the original Diff-instruct’s FID. Notice that a better performance of 10.88 is still achieved with our proposed convergence trajectory, and the improvements also apply to other datasets. We discuss these two mismatch issues in our revised Appendix D.2 and highlight the importance of initializing $s_\phi$.
> >
> > [1] Franceschi et al., Unifying GANs and Score-Based Diffusion as Generative Particle Models, NeurIPS 2023
> >
> > [2] Luo et al., Diff-Instruct: A Universal Approach for Transferring Knowledge From Pre-trained Diffusion Models, NeurIPS 2023
> >
> > [3] Wei et al., Adversarial Score Distillation: When score distillation meets GAN, CVPR 2024
> >
> > **[W2]  Adopting the DMD2 codebase**
> >
> > **Response to W2**
> >
> > The code we provided is the one used for our experiments with SDXL, which is built on the DMD framework. We utilized the proposed DisBack method to distill the SDXL model, producing the results shown in Table 4. Our implementations on ImageNet, FFHQ, and AFHQv2 were based on Diff-instruct with the same dataloader and update rule, but not on the DMD framework. Thus, the comparison of these datasets in Table 1 and 2 are mainly between ours and Diff-Instruct. Our performance gains on ImageNet are attributed to DisBack.

---

> > ### Author Response · Authors · 2024-11-25
> >
> > **[W3]  Configuration files for experiments on FFHQ and ImageNet**
> >
> > **Response to W3**
> >
> > As mentioned in the response to W2, our implementations on FFHQ and ImageNet are built on Diff-instruct, which does not have configuration files. Here we provide the key code of initialize $s_\phi$ and training $G_{stu}$ with convergency trajectory and the command to train the model.
> >
> > ```python
> > # The distribution back-tracking stage
> > while True:
> >     # switch checkpoint along the convergence trajectory
> >     if backtracking_path and cur_tick % switch_gap == 0 and cur_tick != 0 and ckpt_num >=50:
> >         ckpt_num -= 50
> >         net = torch.load(os.path.join(base_backtracking_path, f'intermediate-{ckpt_num}.pth'))['Sg'].to(device)
> >         dist.print0(f'distill to chkpt {ckpt_num}')
> >         if ckpt_num <= 100:
> >             switch_gap = 10000
> >
> >     # To train s_phi
> >     optimizer.zero_grad(set_to_none=True)
> >     # Accumulate gradients.
> >     for round_idx in range(num_accumulation_rounds):
> >         with misc.ddp_sync(Sgddp, (round_idx == num_accumulation_rounds - 1)):
> >             # load training data with Diff-Instruct data loader
> >             images, labels = next(dataset_iterator)
> >             images = images.to(device).to(torch.float32) / 127.5 - 1
> >             labels = labels.to(device)
> >
> >             # sample from the student
> >             with torch.no_grad():
> >                 G.eval()
> >                 z = init_sigma*torch.randn_like(images)
> >                 gen_images = G(z, init_sigma*torch.ones(z.shape[0],1,1,1).to(z.device), labels, augment_labels=torch.zeros(z.shape[0], 9).to(z.device))
> >                 G.train()
> >
> >             # perform distillation
> >             loss = loss_fn(net=Sgddp, images=gen_images, labels=labels, augment_pipe=augment_pipe)
> >             training_stats.report('SgLoss/loss', loss)
> >             loss.sum().mul(sgls / batch_gpu_total).backward()
> >
> >     # Update weights.
> >     if lr_rampup_kimg > 0:
> >         for g in optimizer.param_groups:
> >             g['lr'] = sg_optimizer_kwargs['lr'] * min(cur_nimg / max(lr_rampup_kimg * 1000, 1e-8), 1)
> >
> >     for param in Sg.parameters():
> >         if param.grad is not None:
> >             torch.nan_to_num(param.grad, nan=0, posinf=1e5, neginf=-1e5, out=param.grad)
> >
> >     optimizer.step()
> >
> >     # To train the student
> >     g_optimizer.zero_grad(set_to_none=True)
> >     for round_idx in range(num_accumulation_rounds):
> >         with misc.ddp_sync(Gddp, (round_idx == num_accumulation_rounds - 1)):
> >             # sample from the student
> >             z = init_sigma*torch.randn_like(images)
> >             gen_images = Gddp(z, init_sigma*torch.ones(z.shape[0],1,1,1).to(z.device), labels, augment_labels=torch.zeros(z.shape[0], 9).to(z.device)) #! -1,1
> >             # form loss
> >             Sg.eval()
> >             loss = loss_scaling*loss_fn.gloss(Sd=net, Sg=Sg, images=gen_images, labels=labels, augment_pipe=None)
> >             Sg.train()
> >             loss = loss.sum([1,2,3])
> >             training_stats.report('GLoss/loss', loss)
> >             loss.sum().mul(1.0 / batch_gpu_total).backward()
> >
> >     # Update weights.
> >     if lr_rampup_kimg > 0:
> >         for g in g_optimizer.param_groups:
> >             g['lr'] = g_optimizer_kwargs['lr'] * min(cur_nimg / max(lr_rampup_kimg * 1000, 1e-8), 1)
> >
> >     for param in G.parameters():
> >         if param.grad is not None:
> >             torch.nan_to_num(param.grad, nan=0, posinf=1e5, neginf=-1e5, out=param.grad)
> >
> >     g_optimizer.step()
> >
> > # commands for training on imagnet
> > # torchrun --standalone --nproc_per_node=4 --master_port=25212 di_train.py --outdir=logs --data=datasets/imagenet_edm_64x64.zip --arch=adm --batch 8 --edm_model imagenet64-cond --cond=1 --metrics fid50k_full --tick 10 --snap 100 --lr 0.00001 --glr 0.00001 --init_sigma 1.0 --fp16=0 --lr_warmup_kimg -1 --ls 100.0 --sgls 100.0 --seed 22134 --backtracking_path logs/00011-imagenet_edm_64x64-cond-non_backtracking-ls1.0-sgls1.0-glr1e-05-sglr1e-05-sigma1.0-gpus1-batch8-fp32-lrwarmkimg-1/chkpt/intermediate-200.pth --dropout=0.10 --augment=0
> >
> > # commands for training on ffhq
> > # torchrun --standalone --nproc_per_node=4 --master_port=25212 di_train.py --outdir=logs --data=datasets/ffhq-64x64.zip --arch=ddpmpp --batch 8 --edm_model ffhq64-uncond --cond=0 --metrics fid50k_full --tick 10 --snap 500 --lr 0.00001 --glr 0.00001 --init_sigma 1.0 --fp16=0 --lr_warmup_kimg -1 --ls 100.0 --sgls 100.0 --seed 22134  --cres=1,2,2,2 --backtracking_path logs/00027-ffhq-64x64-ls1.0-sgls1.0-glr1e-05-sglr1e-05-sigma1.0-gpus1-batch16-fp32-lrwarmkimg-1/chkpt/intermediate-200.pth
> > ```
> >
> > To further see the detailed meaning of each parameter in the command, please refer to [di_train.py](https://github.com/pkulwj1994/diff_instruct/blob/main/di_train.py#L60).
> >
> > With less than 3 days remaining until the rebuttal deadline, if you have any questions about our responses or any other concerns regarding our paper, please feel free to comment and let us know. We will do our best to address your concerns promptly.

---

> > > ### Comment · Reviewer_vTDX · 2024-11-29
> > >
> > > Thank you for your detailed responses, which have adequately addressed my concerns. I am therefore increasing my rating to a 6. Please ensure to incorporate all the necessary details and clarifications into the revision.

---

> ### Author Response · Authors · 2024-11-29
> **Thanks for your support**
>
> Please accept our sincerest gratitude for your valuable suggestions and the score improvement for our paper. We have incorporated the necessary details and clarifications into the paper, primarily including the following parts (marked in red in the paper)
>
> 1. We have corrected the total number of training iterations on FFHQ, AFHQv2, and ImageNet datasets to 50k.
> 2. We have further discussed two types of mismatch issues in Appendix D.2 and highlighted the importance of initializing $s_{\phi}$.
> 3. Some other revisions on the expression of the paper.
>
> If you have any other questions or concerns, please feel free to comment and let us know. We will try our best to provide detailed clarifications and explanations to address them promptly.
>
> Thank you once again for your invaluable contribution to our research.

---

### Official Review · Reviewer_yzv7 · 2024-11-04

**Soundness:** 2
**Presentation:** 3
**Contribution:** 3
**Rating:** 8
**Confidence:** 4

**Summary:**

This paper introduces a novel approach to improve diffusion distillation. The key idea is to include a trajectory of teacher distributions for the student to match. This improves convergence and the final quality of the student model.

**Strengths:**

This paper proposes a novel technique that is intuitive and is shown to work well. The experiments show a clear improvement in the convergence speed of the DisBack method and the qualitative results show high-quality single-step samples. The authors apply their method to a variety of teacher models over multiple datasets and demonstrate success and failure cases (in the appendix).

The paper is easy to follow. Technical content is presented clearly.

**Weaknesses:**

I felt that the paper writing in places suggested more than was provided. For example, the authors claim that they "identified this issue arises because existing score distillation methods focus on using the endpoint". However, the authors provide no such identification in their own work. They provided sufficient evidence that utilizing a trajectory of distributions improves the student but this has not been shown to be necessary. There may be alternative approaches that work well while using only the endpoint. This is present elsewhere in the work, e.g. "the fat convergence speed is because constraining the convergence trajectory of the generator provides a clear optimization direction", this is a strong technical claim that has not been adequately explored.

The authors could do more to explain why DisBack is successful. Some theoretical analysis could help to explain the improved convergence speed, or experiments designed to show the convergence more carefully. For instance, I'd be interested to see how quickly the student model converges to each point on the trajectory, and how closely it matches the distribution. Presumably, this behaviour must be better than linear for DisBack to succeed, which is interesting. Overall, I felt that the idea worked well and was intuitive, but I didn't understand why it worked so well after reading the paper.

The ablation study is minimal (and I would argue does not qualify as an ablation study). The authors only explore DisBack and the original method side-by-side. Instead, they could also investigate the effect of the number of degradation path checkpoints, different student initializations, different degradation schemes (like using the training trajectory), etc.

**Questions:**

The original motivation suggests that the teacher training trajectory could be used, but "the convergence trajectory of most teacher models is inaccessible". It's not clear to me that the training trajectory of the teacher would be useful for distillation, as it may not align well with the student distribution either. Did you explore DisBack using the training trajectory for trained diffusion models?

Did you explore distillation into smaller student models? Mismatched architectures could be a useful application of DisBack too.

Do you have any samples from the models along the intermediate teacher trajectory? Do these produce sensible samples at all?

Overall, I like the proposed DisBack algorithm and feel that it is sufficiently novel and performant to justify publication. I would give the paper a higher score if the authors provided some more experimental investigation into why their method is successful.


Minor:

Fig 2. includes the same sample twice (top middle, and middle).

---

> ### Author Response · Authors · 2024-11-19
>
> We truly appreciate your constructive feedback. In light of these insightful comments, we would like to address them by providing the following clarifications and adjustments. To make it easier for the reviewer to understand our response, we pick answer Q1 first.
>
> **[Q1 Using the training trajectory for trained diffusion models.]**
>
> **[Response to Q1]**
>
> First, we clarify that our convergence trajectory refers to the "planned" trajectory from the student towards the teacher model, not the trajectory when training the teacher model originally. The trajectory is useful because it closely guides how the student converges in distillation. Second, our method emphasizes practicality and the proposed degradation approach is designed with feasibility. When pre-trained checkpoints are available, DisBack can be used for distillation. In contrast, using the training trajectory of the teacher would require training from scratch, which would bring huge costs and make our method less practical.
>
> **[W1] How to identify the mismatch issue and why using convergence trajectory is necessary.**
>
> **[Response to W1]**
>
> Identification of the mismatch issue: We give a brief formal explanation and then provide further evidence of the mismatch issue. The pre-trained U-net $s_\theta$ fits the training distribution $q_0$. Because the initial generator's distribution $q^G_0$ differs from $q_0$, the prediction of $s_\theta$ for generated samples with noise is unreliable. To validate this, we conduct further experiments in Appndx E.1. The mismatch degree defined in Eq.(9) (p8) indicates whether the predicted optimization direction for the generator matches the approximated score, and it derives from the score matching loss. The lower the better. When using a pre-trained EDM model (the endpoint) on ImgNet and FFHQ, we calculated the mismatch degree on both samples of the initial student model’s $G_{stu}^0$ and the teacher model’s training samples. Results show the degree of the generated samples is higher than that of the training. This means the predicted direction from the endpoint deviates from the correct direction. Therefore, the mismatch issue is identified. Additionally, we compared the mismatch degree during training in our DisBack and Diff-instruct (Luo et al., 2023c) which uses the endpoint only. As in Fig 4 in Sec 5.3 (p8), our method shows a lower mismatch degree, therefore allowing the student to enjoy a clear optimization direction and thus fast convergence. Hence, only using the endpoint is not enough, and utilizing the convergence trajectory beyond the endpoint is necessary.
>
> **[W2] Why DisBack is successful.**
>
> **[Response to W2]**
>
> The reason for DisBack's success. We give an extra brief theoretical explanation and introduce related experiments. In the degradation stage, we construct a series of diffusion models $\lbrace s_{\theta_i}' \mid i=0, \ldots, N\rbrace$ where $s_{\theta_0}' = s_\theta \approx q_0$ and $s_{\theta_N}'  \approx q^G_0$. The distribution mismatch issue is explained by $s_{\theta_0}'  \neq s_{\theta_N}'$. Furthermore, notice that with a large $i$,$s_{\theta_i}'$is closer to $s_{\theta_N}'$, while with a small $i$,$s_{\theta_i}' $is far away from $s_{\theta_N}'$but close to $s_{\theta_0}'$. Therefore, given samples from $s_{\theta_N}'$, as they are closer to $s_{\theta_{N-1}}'$than to $s_{\theta_0}'$, the predicted scores given by $s_{\theta_{N-1}}'$ are more accurate than that by $s_{\theta_0}'$. Given more accurate scores, the student model enjoys improved convergence speed, as in our response to W1.
> Our Fig 1 and 4 visualize experiments designed to show DisBack's advantage in convergence. Fig 1 displays how quickly the student model converges to the first two checkpoints on the trajectory.
> |Method|$s_{\theta_4}'$|$s_{\theta_3}'$|$s_{\theta_2}'$|$s_{\theta_1}'$|$s_{\theta_0}'$|
> |---|---|---|---|---|---|
> |DisBack|21.58|15.48|11.81|7.43|1.38|
> |Diff-Instruct|71.12|22.13|12.35|9.19|5.96
>
> The above Tab further shows the FID values of our DisBack when it reaches each checkpoint and the values when Diff-Instruct trains the same number of items on ImgNet64. DisBack converges to each point more quickly. Experiments in Fig 4 (p8) show how closely the student matches the distribution as discussed in our response to W1. In summary, the reason for the efficacy of our method lies in that incorporating the convergence trajectory effectively alleviates the distribution mismatch problem between the teacher and student models, thereby accelerating the overall convergence process.

---

> ### Author Response · Authors · 2024-11-19
>
> **[W3] The additional ablation studies.**
>
> **[Response to W3]**
>
> We are working diligently to complete this ablation experiment, and the results will be provided during the rebuttal period.
>
>
> **[Q2] The distillation to smaller student models.**
>
> **[Response to Q2]**
>
> We explore this case by attempting to distill a pre-trained EDM model into a FastGAN generator. Our DisBack outperforms the original FastGAN, the EDM model with 11 NFE and ScoreGAN (Appndx A.2).
>
>
> **[Q3] Samples from from the models along the intermediate teacher trajectory.**
>
> **[Response to Q3]**
>
> We visualized the generated images of intermediate checkpoints originally trained on ImgNet64 (Appndx A.3). As the degradation process, the images gradually became chaotic and blurred, eventually approaching the generated images of the student model. However, early samples are sensible so the early ckpts are still able to guide the student.
>
>
> **[Q4]  The same sample in Fig.2**
>
> **[Response to Q4]**
>
> Thank you very much for pointing out this. We have made the corresponding revisions in the paper.

---

> ### Comment · Reviewer_yzv7 · 2024-11-21
>
> Thank you for your response. Several points have been adequately clarified for me, and I appreciate the new results that you have included.
>
> I am happy with the clarifications and responses provided for Q1, W3 [this can be included in the final version and need not be rushed for rebuttal], Q2, Q3, and Q4.
>
> I still have some lingering questions regarding W1 and W2. I want to clarify also that while your response addresses what is happening my questions are primarily about **why** it is happening. I hope I can make this clearer in this response.
>
> [W1] I understand that the student generator will have a worse match to the true data distribution than the teacher. In some sense this is obvious --- nobody is disputing that the 1-step teacher model has a different distribution than the N-step teacher model. The part of your response that I cannot follow is "the predicted direction from the endpoint deviates from the correct direction.". The mismatch degree measures the average difference in the teacher score and the true score on some dataset. But when optimizing the student we do not move along this score function --- it tells us nothing directly about how easy it is to optimize the student.
>
> I acknowledge that the mismatch degree shows that DisBack achieves a closer match to the teacher than Diff-Instruct --- this is a nice result. But for me, the question of why remains. Please do correct me if I have misunderstood any of the above.
>
> [W2] My original comment was not particularly clear but I feel that your response has largely addressed my concerns here. However, I do want to clarify what I meant.
>
> As DisBack requires $N$ separate optimization problems to be solved, for accelerated convergence the solution of each problem must be found more quickly than $T/N$ where $T$ is the time to solve the original Diff-Instruct distillation. I am mostly interested in why this happens and under what conditions. My proposed experiment was to evaluate distillation of $q_{0}^{G}$ to each of the $s_{\theta_{i}}$ independently (not sequentially).
>
> This set of experiments would include the original Diff-Instruct problem (where we optimize against the teacher endpoint), and the first DisBack optimization problem. But we'd also see how the convergence speeds degrades on the distillation problems along the trajectory. This would help identify how many degradation checkpoints we need to maintain fast convergence and how quickly the convergence speed falls off.
>
> Related to these points, is the fact that Diff-Instruct (and similar methods) also optimize a score estimator for the student distribution. Why is this insufficient to capture the mismatched distribution but DisBack works?
>
> Summarily, I think ample evidence has been provided that DisBack (a) improves the convergence speed of distillation relative to using the teacher alone and (b) improves the final quality of the distilled model. But I remain unconvinced that the authors have demonstrated convincingly why this happens, and would still prefer that some of the claims highlighted in my original review are either toned down or addressed directly.
>
> I'd also like to reiterate that I think this method is a valuable contribution with a lot of utility and I am increasing my score to reflect that I believe it should be published.

---

> > ### Author Response · Authors · 2024-11-22
> >
> > We would like to express our greatest appreciation for the increased score. We also thank the helpful comment with suggested experiments. We provide more explanation and quick results as follows.
> >
> > **[W1] Why the mismatch happens.**
> >
> > **Response to W1**
> >
> > The reason why the predicted direction $s_\theta(x_t,t)$ from the endpoint deviates from the true direction $\nabla_{x_t} \log q_t(x_t)$ ($\nabla_{x_t} \log p_t(x_t)$ in previous Eq. 9 is a typo) is that the initially generated sample $x_0 = G_{stu}^0({z};\eta)$ is out of the distribution of the pre-trained diffusion model $q_0$ and thus $s_\theta(x_t,t)$ is unreliable in this case.
> > Recall that the training data are real images of the pre-trained model subject to $q_0$， and $q_t$ is the noisy distribution at timestep $t$. The pretrained $s_\theta$ is only trained on $q_t$ and it approximates $\nabla_{x_t} \log q_t(x_t)$(Ln 207-208).  As shown in Eq.(4) (Ln 205 in p4,), the gradient given to $G_{stu}^0$is $\left[ \nabla_{x_t}\log q^G_{t}\left(x_t\right) - \nabla_{x_t} \log q_{t}\left(x_t\right)\right] \frac{\partial x_t}{\partial \eta}$, which is approximated by $\left[s_{\phi}\left(x_t, t\right)-s_{\theta}\left(x_t, t\right)\right]\frac{\partial x_t}{\partial \eta}$ (Eq.(4), p4, Ln 215) . Given a noisy sample $x_t$, the direction of $s_{\phi}\left(x_{t}, t\right)$ points to the generated distribution and the direction of $s_{\theta}\left(x_t, t\right)$ points to the training distribution. Ideally, if the predicted score of $s_{\theta}\left(x_t,t \right)$is correct, the direction of the gradient is from the generated distribution towards the training distribution, and this gradient leads $G_{stu}^0$ to update in the right direction. We discuss this point in Sec.D.1 (Fig.10) and verify the direction of the gradient in 2 dimension data in Sec.E.3 (Fig 13).
> >
> > Simultaneously,  an initially generated sample $x_0 = G_\mathit{stu}^0({z};\eta) \sim q_G^0$ is an artefact image outside the support of $q_0$. Also, $q_G^0$ is "far away" from $q_0$. The noisy sample $x_t$ of $x_0$ also tends to lie outside the distribution of $q_t$, especially when $t$ is small. In this case, the prediction $s_\theta(x_t,t)$ is unreliable and fails to approximate the true $\nabla_{x_t} \log q_t(x_t)$. Thus, the gradient $\left[s_{\phi}\left(x_t, t\right)-s_{\theta}\left(x_t, t\right)\right]\frac{\partial x_t}{\partial \eta}$ is unreliable and the student cannot get reliable guidance. This is the mismatch issue we focus on this paper and the mismatch degree is defined as the difference between $s_\theta(x_t,t)$ and $\nabla_{x_t} \log q_t(x_t)$ over generated samples or real samples (Eq. 9 and Fig. 4). Specifically, for a single sample $x_t$, the difference between $s_\theta(x_t,t)$ and  $\nabla_{x_t} \log q_t(x_t)$ may reflect the degree that ${x}_{t}$ is outside $q_t$. Such difference is relatively large for Diff-Instruct.
> >
> > On the other hand,$s_{\theta_i}'(x_t,t)$, an intermediate ckpt along the convergence trajectory, represents non-existent distributions $q_0^{(i)}$ closer to $q^G_0$ than $q_0$. Therefore, the noisy generated sample $x_t$lies closer to $q_0^{(i)}$than to $q_0$. The prediction $s_{\theta_i}'(x_t,t)$ for $x_t$ is closer to  $\nabla_{x_t} \log q_t(x_t)$and thus more reliable. The following table shows the augmented mismatch degree when $s_\theta$is replaced by other intermediate ckpts  as $s_{\theta_4}$, $s_{\theta_3}$, $s_{\theta_2}$and $s_{\theta_1}$and $x_t$ are the initially generated samples. The first ckpt $s_{\theta_4}$enjoys the lowest mismatch degree in this case.
> >
> > |$s_{\theta_4}$|$s_{\theta_3}$|$s_{\theta_2}$|$s_{\theta_1}$|$s_{\theta_0}$|
> > |-|-|-|-|-|
> > |0.166|0.174|0.199|0.235|0.288|

---

> > ### Author Response · Authors · 2024-11-22
> >
> > **[W2.1] Why the acceleration happens and under what conditions**
> >
> > **Response to W2.1**
> >
> > We have explained why the mismatch happens in our response in W1. Hence for the why part, as the mismatch is alleviated and the guidance is accurate, the distillation is accelerated.
> > Accordingly, acceleration happens when given two situations which often occurs. Firstly, the initial$q_0^G$is "far away" from the teacher distribution represented by$s_\theta$, especially at the early stage of distillation. Secondly, when $t$ is small, $x_t$ remains more features of the original generated images with a smaller noise scale, making it more likely to fall out of the distribution of $s_\theta$. As the noise scale increases, $x_t$ approaches standard Gaussian and tends to lie within the distribution of $s_\theta$.
> >
> > **[W2.2] To evaluate distillation of $q^G_0$ to each of the $s_{\theta_i}$ independently**
> >
> > **Response to W2.2**
> >
> > We conduct your proposed experiment and use our five checkpoints from the degradation path as teacher models to distill the generator independently. The table below shows their resulting FID values at 1000 steps. This shows how the convergence speed degrades along the trajectory. The student distillated with $s_{\theta_4}$ gets the lowest FID in this case. The convergence speed falls from $s_{\theta_4}$ to $s_{\theta_0}$ because the mismatch is more significant.
> >
> > |$s_{\theta_4}$|$s_{\theta_3}$|$s_{\theta_2}$|$s_{\theta_1}$|$s_{\theta_0}$|
> > |-|-|-|-|-|
> > |21.58|26.19|32.22|49.45|71.12|
> >
> > Notice that the accelerated is not uniformly distributed as smaller T/N. Each ckpt may have a different acceleration effect, and the endpoint $s_{\theta_0}$does not accelerate the process.
> >
> > **[W2.3] Why the score estimator for the student distribution is insufficient to capture the mismatched distribution**
> >
> > **Response to W2.3**
> >
> > We think the referred score estimator for the student distribution is $s_\phi$as trained with Eq (5) and used in Eq (6), and we agree $s_\phi$ is sufficient to capture the student distribution. We guess the reviewer thinks the discussed mismatched distributions are the difference between $s_\phi$ and $s_\theta$. In fact, this is to be solved by the distillation methods. The mismatch issue we discuss is the mismatch between $s_\theta(x_t, t)$ and $\nabla_{x_t} \log q_t(x_t)$ , irrelevant to $s_\phi$. DisBack shares the same $s_\phi$ with Diff-Instuct, but it works because it uses $s_{\theta_i}(x_t, t)$, a reliable guidance closer to $\nabla_{x_t} \log q_t(x_t)$ than $s_\theta(x_t, t)$.

---

> ### Comment · Reviewer_yzv7 · 2024-11-25
>
> I greatly appreciate your patience and feel that I now better understand the theoretical justification. I will provide some more feedback here in the hopes that it can help to clarify things for other readers.
>
> Indeed, the unfortunate typo $p_t$ was causing me some confusion [note that this typo still exists in the body of text describing Equation 9]. Moreover, I had failed to connect that the "assessed distribution" of equation 9 is the noisy samples from the student output (as used in Diff-Instruct training). Now that I understand what is meant, the writing does make sense. I now interpret the mismatch as measuring the difference between the teacher's score estimate and the true score under the data distribution. Which when connected to Equations 4 and 6, highlight why difficulty in optimization may arise.
>
> I would suggest including some of the details presented above within Section 5.3 to help clarify this for other readers who might suffer similar misconceptions. Here are some changes I would consider:
>
> - Include distribution notation for the assessed distribution in Equation 9. For example, $d_{mis}(f, s\_\theta) = \mathbb{E}\_{x\_0 \sim f} \mathbb{E}\_{x\_t \sim N(x\_0, \sigma\_t)}\left[ s\_\theta(x\_t, t) - \nabla\_{x\_t} \log q\_t (x\_t)\vert x\_0 \right]$
> - Explicitly point to Equations 4 and 6 to highlight why this is a reasonable metric for measuring training difficulty.
>
> On W2.3, my misunderstanding was assuming that the $s_\phi$ addresses a mismatch between the true data distribution and the student distribution. However, the goal instead is to address the mismatch between the teacher's estimate of the true data distribution and the true data distribution when fed the student generator's output.
>
> Also, another very minor point. I think the notation in Section 3 $p(x\_t | x\_0) \sim N (x\_0, σ^2\_t)$ isn't valid. It should instead be $x\_t | x\_0 \sim N (x\_0, σ^2\_t)$.

---

> > ### Author Response · Authors · 2024-11-25
> >
> > Please accept our heartfelt gratitude once again for providing such valuable suggestions on our paper. We have incorporated the suggested modifications into the manuscript. Additionally, we have included the results of the ablation study mentioned in W3.
> >
> > **[W3] The additional ablation studies.**
> >
> > **Response to W3**
> >
> > We have completed the experiment on different $N$, the number of checkpoints on the degradation path. In our default setting, the DisBack degradation process involved 200 iterations, with $N=5$ checkpoints in the path (saving one checkpoint every 50 iterations). We evaluated the performance on the ImageNet64 dataset for $N=3$ (saving one checkpoint every 100 iterations) and  $N=11$  (saving one checkpoint every 20 iterations).  Due to time limitations, we were unable to conduct experiments for  N=2  and  N=4. Additionally, we extended the degradation process to 400 iterations and evaluated the performance for  $N=11$ (saving one checkpoint every 40 iterations).  The results are presented in Appendix G of the revised paper.
> >
> > |N=1|N=3|N=5 (default setting) |N=11|N=11 (Degradation iteration=400)|
> > |-|-|-|-|-|
> > |5.96|4.88|1.38|9.15|244.72|
> >
> > For $N=3$, DisBack achieved an FID of 4.88, while for  $N=11$, the FID increased to 9.15.  The performance degradation at  $N=11$ results from that when there are too many checkpoints in the path, the distributions of certain checkpoints (e.g.,  $s_{\theta_{10}}^\prime, s_{\theta_{9}}^\prime, s_{\theta_{8}}^\prime$) are very close to the distribution of the initial generator.  Training with these checkpoints provides limited progress for the generator.  Furthermore, having too many checkpoints complicates the checkpoint transition scheduler during distillation, making it difficult to manage effectively.  This often results in inefficient updates to the generator across many iterations, wasting time without achieving meaningful improvements. When the degradation iteration count is set to 400, we observed that the student model could not be effectively trained.  This is because excessive degradation iterations lead to checkpoints near the initial generator on the degradation path being unable to generate reasonable samples.  Using these checkpoints for distillation fails to provide the generator with meaningful or effective guidance, ultimately resulting in the generator’s inability to move quickly towards the training distribution within limited iteration.
> >
> > Therefore, when obtaining the degradation path, it is crucial not to over-degrade the teacher model.
> > We agree there is another balance concerned about the number of degradation iterations. If $s_{\theta_{N}}^\prime$ is too far away from $q_G^0$, our discussed mismatch issue arises. Simultaneous, if $s_{\theta_{N}}^\prime$ is too close to  $q^G_0$, $s_{\theta_{N}}^\prime$degrades too much and fails to give useful guidance to the student. In our extensive experiments on 5 datasets, we empirically find using our default setting (5 checkpoints in 200 degradation iterations) is likely to result in improved performance.

---

### Author Response · Authors · 2024-11-29
**Global Response**

We sincerely appreciate all the reviewers for their thorough examination and valuable suggestions. We are glad to hear that the proposed idea is intuitive, effective and novel (Reviewers vTDX, JLT9 and yzv7), the proposed method is easy to understand (Reviewers yzv7 and JLT9) and versatile (Reviewers HUTP and vTDX). We have revised the manuscript according to the comments of the reviewers (**highlighted in red**).

Here, we summarize and highlight our responses to the reviewers:
* We added a preliminary experiment to provide evidence for the score mismatch issue present in existing score distillation methods (Reviewer yzv7) and further conducted a detailed analysis of the motivation and effectiveness of the proposed DisBack method intuitively (Reviewers yzv7 and HUTP).
* We included an ablation study on the number of checkpoints and different degradation schemes to further explain and validate the effectiveness of DisBack (Reviewers yzv7, JLT9 and HUTP).
* We added a discussion on different types of mismatch issues (Reviewer vTDX) and a comparison with existing related work (Reviewer JLT9).
* We revised Fig. 1 to compare DisBack with Diff-instruct while accounting for the computational cost of the first stage (Reviewer vTDX, JLT9 and HUTP).
* We revised the unclear statements in the paper to avoid confusion (Reviewers vTDX, yzv7, and JLT9).

We reply to each reviewer's questions in detail below their reviews. Please kindly check out them. Thank you and please feel free to ask any further questions.

---

### Meta-Review · Area_Chair_U4Kf · 2024-12-24

**Metareview:**

This paper targets enhancing diffusion distillation through a novel means of resolving the score mismatch issue by incorporating a trajectory of teacher distributions for student matching, which boosts convergence and student model quality. Reviewers generally find the method novel and captivating, with significant experimental outcomes. The experiments evidently enhance DisBack's convergence speed, and qualitative results present high-quality single-step samples. However, concerns pertain to justifying experimental results and analyzing the sensitivity of backtracking schedule choices. AC agrees this is a novel contribution to the community, hence recommending it as a poster.

**Additional Comments On Reviewer Discussion:**

The principal concerns articulated by the reviewers are as follows:

1) The methodology for identifying the mismatch issue and the rationale behind the necessity of using the convergence trajectory require clarification.
2) An explanation is needed as to why the DisBack approach is successful.
3) The basis for the student model's ability to outperform the teacher model demands elucidation.

Following the rebuttal, the majority of issues and misunderstandings have been satisfactorily addressed. However, one reviewer persists in believing that concern 3 persists.

In my opinion, the authors have performed admirably during the rebuttal process. The contribution presented in the current submission version is of interest to the community and is adequate for publication.

---

### Decision · Program_Chairs · 2025-01-22

Accept (Poster)